# DIRECTLY FINE-TUNING DIFFUSION MODELS ON DIFFERENTIABLE REWARDS

**Kevin Clark**\*, **Paul Vicol**\*, **Kevin Swersky, David J. Fleet**
Google DeepMind      \*Equal contribution
{kevclark, paulvicol, kswersky, davidfleet}@google.com

## ABSTRACT

We present Direct Reward Fine-Tuning (DRaFT), a simple and effective method for fine-tuning diffusion models to maximize differentiable reward functions, such as scores from human preference models. We first show that it is possible to back-propagate the reward function gradient through the full sampling procedure, and that doing so achieves strong performance on a variety of rewards, outperforming reinforcement learning-based approaches. We then propose more efficient variants of DRaFT: DRaFT-$K$, which truncates backpropagation to only the last $K$ steps of sampling, and DRaFT-LV, which obtains lower-variance gradient estimates for the case when $K = 1$. We show that our methods work well for a variety of reward functions and can be used to substantially improve the aesthetic quality of images generated by Stable Diffusion 1.4. Finally, we draw connections between our approach and prior work, providing a unifying perspective on the design space of gradient-based fine-tuning algorithms.

## 1 INTRODUCTION

Diffusion models (Sohl-Dickstein et al., 2015; Song & Ermon, 2019; Ho et al., 2020; Song et al., 2021b; Kingma et al., 2021) have revolutionized generative modeling for continuous data, achieving impressive results across modalities including images, videos, and audio. However, for many use cases, modeling the training data distribution exactly (e.g. diverse images from the web) does not align with the model's desired behavior (e.g. generating aesthetically pleasing images). To overcome this mismatch, text-to-image diffusion models commonly employ methods like classifier-free guidance (Ho & Salimans, 2021) to improve image-text alignment, and are fine-tuned on curated datasets such as LAION Aesthetics (Schuhmann & Beaumont, 2022) to improve image quality.

Within the language modeling community, aligning model behavior with human preferences is widely accomplished through fine-tuning with reinforcement learning (RL), such as with RL from human feedback (RLHF; Ziegler et al. 2019; Stiennon et al. 2020; Ouyang et al. 2022; Bai et al. 2022a). Inspired by RLHF, supervised and reinforcement learning methods have been developed to train

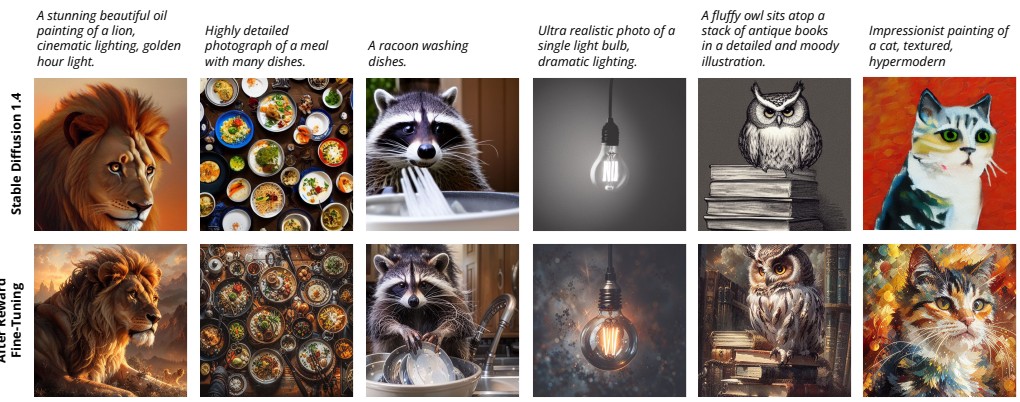

*A stunning beautiful oil painting of a lion, cinematic lighting, golden hour light.* | *Highly detailed photograph of a meal with many dishes.* | *A racoon washing dishes.* | *Ultra realistic photo of a single light bulb, dramatic lighting.* | *A fluffy owl sits atop a stack of antique books in a detailed and moody illustration.* | *Impressionist painting of a cat, textured, hypermodern*

Figure 1: DRaFT fine-tuning for human preference reward models yields more detailed and stylized images than baseline Stable Diffusion. See Figure 37 in Appendix E for more samples.

diffusion models on human preference models or other rewards (Lee et al., 2023; Wu et al., 2023b; Dong et al., 2023; Black et al., 2023; Fan et al., 2023). While showing some success at improving text alignment or image aesthetics on narrow domains, these approaches are sample inefficient and have not been scaled up to the level of generally improving generation style across diverse prompts.

We propose a simple and efficient approach for gradient-based reward fine-tuning based on differentiating through the diffusion sampling process. We first introduce Direct Reward Fine-Tuning (DRaFT), a conceptually straightforward method that backpropagates the reward through the full sampling chain. To keep memory and compute costs low, we use gradient checkpointing (Chen et al., 2016) and optimize LoRA weights (Hu et al., 2022) rather than the full set of model parameters. Then, we introduce modifications to DRaFT that further improve its efficiency and performance. First, we propose DRaFT-$K$, a variant that backpropagates through only the last $K$ steps of sampling to compute the gradient. We show empirically that full backpropagation can result in exploding gradients, and that using the truncated gradient performs substantially better given the same number of training steps. Then, we further improve efficiency by introducing DRaFT-LV, a variant of DRaFT-1 that computes lower-variance gradient estimates by averaging over multiple noise samples.

We apply DRaFT to Stable Diffusion 1.4 (Rombach et al., 2022) and evaluate it on a variety of reward functions and prompt sets. As our methods leverage gradients, they are substantially more efficient than RL-based fine-tuning baselines, for example maximizing scores from the LAION Aesthetics Classifier (Schuhmann & Beaumont, 2022) $>200\times$ faster than the RL algorithms from Black et al. (2023). DRaFT-LV is particularly efficient, learning roughly $2\times$ faster than ReFL (Xu et al., 2023), a previous gradient-based fine-tuning method. We scale up DRaFT to train on hundreds of thousands of prompts and show that it improves Stable Diffusion on human preference rewards such as PickScore (Kirstain et al., 2023) and Human Preference Score v2 (Wu et al., 2023a). We further show that we can interpolate between the pre-trained and fine-tuned models by scaling the LoRA weights, and we can combine the effects of multiple rewards by linearly combining fine-tuned LoRA weights. Finally, we showcase how DRaFT can be applied to diverse reward functions including image compressibility and incompressibility, object detection and removal, and generating adversarial examples.

## 2 RELATED WORK

We provide an extended discussion of related work in Appendix D.

**Learning human preferences.** Human preference learning trains models on judgements of which behaviors people prefer, rather than on human demonstrations directly (Knox & Stone, 2009; Akrour et al., 2011). This training is commonly done by learning a reward model reflecting human preferences and then learning a policy that maximizes the reward (Christiano et al., 2017; Ibarz et al., 2018). We apply DRaFT to optimize scores from existing preference models, such as PickScore (Kirstain et al., 2023) and Human Preference Score v2 (Wu et al., 2023a), which are trained on human judgements between pairs of images generated by diffusion models for the same prompt.

**Guidance** (Song et al., 2021b; Dhariwal & Nichol, 2021) steers sampling towards images that satisfy a desired objective by adding an auxiliary term to the score function. Various pre-trained models can be used to provide guidance signals, including classifiers, facial recognition models, and object detectors (Bansal et al., 2023). Noisy examples obtained during sampling may be out-of-distribution for the guidance model. Two approaches to mitigate this are: (1) training the guidance model on noisy data (Dhariwal & Nichol, 2021) and (2) applying a guidance model to predicted clean images from one-step denoising (Li et al., 2022). However, (1) precludes the use of off-the-shelf pre-trained guidance functions and (2) applies guidance to out-of-distribution blurry images for high noise levels. DRaFT avoids these issues by only applying the reward function to the final generated image.

**Backpropagation through diffusion sampling.** Watson et al. (2022) differentiate through sampling to learn sampler hyperparameters (rather than model parameters) in order to reduce inference cost while maintaining perceptual image quality. Fan & Lee (2023) also propose an approach to speed up sampling: they consider doing this by backpropagating through diffusion sampling, but focus on an RL-based approach due to memory and exploding/vanishing gradient concerns. Both Fan & Lee (2023) and Watson et al. (2022) aim to improve sampling speed, rather than optimize arbitrary differentiable rewards, which is our focus. Like DRaFT, Direct Optimization of Diffusion Latents (DOODL; Wallace et al. 2023) backpropagates through sampling to optimize differentiable rewards; however DOODL optimizes the *initial noise sample* $\mathbf{x}_T$ rather than the model parameters. While this does not require a training phase, it is much slower at inference time because the optimization

must be redone for each prompt and metric. Outside of image generation, Wang et al. (2023a) train diffusion model policies for reinforcement learning by backpropagating through sampling.

**Reward Fine-Tuning with Supervised Learning.** Lee et al. (2023) and Wu et al. (2023b) use supervised approaches to fine-tune diffusion models on rewards. These methods generate images with the pre-trained model and then fine-tune on the images while weighting examples according to the reward function or discarding low-reward examples. Unlike DRaFT or RL methods, the model is not trained online on examples generated by the current policy. However, Dong et al. (2023) use an online version of this approach where examples are re-generated over multiple rounds of training, which can be viewed as a simple kind of reinforcement learning.

**Reward Fine-Tuning with Reinforcement Learning.** Fan & Lee (2023) interpret the denoising process as a multi-step decision-making task and use policy gradient algorithms to fine-tune diffusion samplers. Building on it, Black et al. (2023) and Fan et al. (2023) use policy gradient algorithms to fine-tune diffusion models for arbitrary black-box objectives. Rather than optimizing model parameters, Hao et al. (2022) apply RL to improve the input prompts. RL approaches are flexible because they do not require differentiable rewards. However, in practice many reward functions are differentiable, or can be implemented or approximated in a differentiable way, and thus analytic gradients are often available. In such cases, using reinforcement learning discards useful information.

**Reward Feedback Learning (ReFL).** ReFL (Xu et al., 2023) uses reward function gradients to fine-tune a diffusion model. It evaluates the reward on the one-step predicted clean image, $r(\hat{\mathbf{x}}_0, \mathbf{c})$ from a randomly-chosen step $t$ along the denoising trajectory rather than on the final image, as DRaFT does. While ReFL performs similarly to DRaFT-1, DRaFT-LV is substantially more efficient, training approximately $2\times$ faster. We further discuss how ReFL relates to DRaFT in Section 4.

## 3 BACKGROUND ON DIFFUSION MODELS

**Diffusion Models.** Diffusion models are latent variable models based on the principle of iterative denoising: samples are generated starting from pure noise $\mathbf{x}_T \sim \mathcal{N}(\mathbf{0}, \mathbf{I})$ via a sequence of applications of a learned denoising function $\boldsymbol{\epsilon}_{\boldsymbol{\theta}}$ that gradually removes noise over $T$ timesteps. Diffusion models often learn the denoiser $\boldsymbol{\epsilon}_{\boldsymbol{\theta}}$ by minimizing the following re-weighted variational lower bound of the marginal likelihood (Ho et al., 2020):

$$\mathcal{L}_{\text{Simple}}(\boldsymbol{\theta}) = \mathbb{E}_{t \sim U(0,T), \mathbf{x}_0 \sim p_{\text{data}}, \boldsymbol{\epsilon} \sim \mathcal{N}(\mathbf{0}, \mathbf{I})} \left[ \| \boldsymbol{\epsilon} - \boldsymbol{\epsilon}_{\boldsymbol{\theta}} \left( \alpha_t \mathbf{x}_0 + \sigma_t \boldsymbol{\epsilon}, t \right) \|^2 \right] \tag{1}$$

where $\sigma_t = \sqrt{1 - \alpha_t^2}$ is an increasing noise schedule.[1] Here, $\boldsymbol{\epsilon}_{\boldsymbol{\theta}}$ is trained to predict the noise $\boldsymbol{\epsilon}$ added to the clean datapoint $\mathbf{x}_0$. For image data, $\boldsymbol{\epsilon}_{\boldsymbol{\theta}}$ is typically parameterized by a UNet (Ronneberger et al., 2015). We use conditional diffusion models, which include a context $\mathbf{c}$ such as a text prompt passed to the denoising function $\boldsymbol{\epsilon}_{\boldsymbol{\theta}}(\mathbf{x}_t, \mathbf{c}, t)$. For inference (i.e., sampling), one draws a noise sample $\mathbf{x}_T$, and then iteratively uses $\boldsymbol{\epsilon}_{\boldsymbol{\theta}}$ to estimate the noise and compute the next latent sample $\mathbf{x}_{t-1}$.

**Classifier-Free Guidance.** Classifier-free guidance (CFG) uses a linear combination of the conditional and unconditional score estimates, denoising according to $(1+w(t))\boldsymbol{\epsilon}(\mathbf{x}_t, \mathbf{c}, t) - w(t)\boldsymbol{\epsilon}(\mathbf{x}_t, \varnothing, t)$ where $w(t)$ is the guidance weight and $\varnothing$ indicates an empty conditioning signal. It requires that $\mathbf{c}$ is replaced with $\varnothing$ part of the time during training so that the unconditional score function is learned.

## 4 METHOD

We propose a simple approach for fine-tuning diffusion models for differentiable reward functions. Our goal is to fine-tune the parameters $\boldsymbol{\theta}$ of a pre-trained diffusion model such that images generated by the sampling process maximize a differentiable reward function $r$:

$$J(\boldsymbol{\theta}) = \mathbb{E}_{\mathbf{c} \sim p_{\mathbf{c}}, \mathbf{x}_T \sim \mathcal{N}(\mathbf{0}, \mathbf{I})} \left[ r(\text{sample}(\boldsymbol{\theta}, \mathbf{c}, \mathbf{x}_T), \mathbf{c}) \right] \tag{2}$$

where $\text{sample}(\boldsymbol{\theta}, \mathbf{c}, \mathbf{x}_T)$ denotes the sampling process from time $t = T \to 0$ with context $\mathbf{c}$.

**DRaFT.** First, we consider solving Eq. 2 by computing $\nabla_{\boldsymbol{\theta}} r(\text{sample}(\boldsymbol{\theta}, \mathbf{c}, \mathbf{x}_T), \mathbf{c})$ and using gradient ascent. Computing this gradient requires backpropagation through multiple diffusion model calls in the sampling chain, similar to backpropagation through time in a recurrent neural network. We use two approaches to reduce the memory cost of DRaFT: 1) low-rank adaptation (LoRA) (Hu et al., 2022); and 2) gradient checkpointing (Chen et al., 2016).

---

[1]We use VDM notation (Kingma et al., 2021), where $\alpha_t$ corresponds to $\sqrt{\bar{\alpha}_t}$ in the Ho et al. (2020) notation.

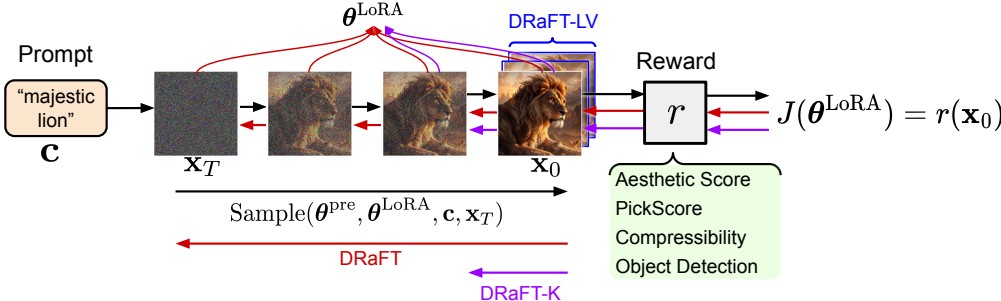

Figure 2: DRaFT backpropagates reward gradients through the sampling process into LoRA weights. DRaFT-$K$ truncates the backward pass, differentiating through only the last $K$ steps of sampling. DRaFT-LV improves the efficiency of DRaFT-1 by generating multiple examples in the last step to reduce variance.

**LoRA.** Rather than fine-tuning the full set of model parameters, Low-Rank Adaptation (LoRA) (Hu et al., 2022) freezes the weights of the pre-trained model, and injects new low-rank weight matrices alongside the original model weights, whose contributions are summed to produce the adapted model outputs. Mathematically, for a layer with base parameters $\mathbf{W}_0$ whose forward pass yields $\mathbf{h} = \mathbf{W}_0\mathbf{x}$, the LoRA adapted layer is $\mathbf{h} = \mathbf{W}_0\mathbf{x} + \mathbf{BA}\mathbf{x}$, where $\mathbf{BA}$ is a low-rank matrix. LoRA dramatically reduces the number of parameters that need to be optimized, which reduces the memory requirements of fine-tuning. Another benefit is that, because the information learned by the fine-tuned model is contained within the LoRA parameters, we can conveniently combine fine-tuned models by taking linear combinations of the LoRA parameters (Figure 5, right) and interpolate between the original and fine-tuned model by re-scaling the LoRA parameters (Figure 7).

---

**Algorithm 1** DRaFT (with DDIM sampling)

---

**Inputs:** pre-trained diffusion model weights $\boldsymbol{\theta}$, reward $r$, prompt dataset $p_{\mathbf{c}}$
**while** not converged **do**

$$t_{\text{truncate}} = \begin{cases} \text{randint}(1,m) & \textbf{if ReFL} \\ T & \textbf{if DRaFT} \\ K & \textbf{if DRaFT-}K \\ 1 & \textbf{if DRaFT-LV} \end{cases}$$

    $\mathbf{c} \sim p_{\mathbf{c}}, \mathbf{x}_T \sim \mathcal{N}(\mathbf{0}, \mathbf{I})$
    **for** $t = T, \ldots, 1$ **do**
        **if** $t = t_{\text{truncate}}$ **then**
            $\mathbf{x}_t = \texttt{stop\_grad}(\mathbf{x}_t)$
        $\hat{\mathbf{x}}_0 = (\mathbf{x}_t - \sigma_t\boldsymbol{\epsilon}_{\boldsymbol{\theta}}(\mathbf{x}_t, \mathbf{c}, t))/\alpha_t$
        **if** $t = t_{\text{truncate}}$ and ReFL **then**
            $\mathbf{x}_0 \approx \hat{\mathbf{x}}_0$
            **break**
        $\mathbf{x}_{t-1} = \alpha_{t-1}\hat{\mathbf{x}}_0 + \sigma_{t-1}\boldsymbol{\epsilon}_{\boldsymbol{\theta}}(\mathbf{x}_t, \mathbf{c}, t)$
    $\boldsymbol{g} = \nabla_{\boldsymbol{\theta}} r(\mathbf{x}_0, \mathbf{c})$
    **if** DRaFT-LV **then**
        **for** $i = 1, \ldots, n$ **do**
            $\boldsymbol{\epsilon} \sim \mathcal{N}(\mathbf{0}, \mathbf{I})$
            $\mathbf{x}_1 = \alpha_1\texttt{stop\_grad}(\mathbf{x}_0) + \sigma_1\boldsymbol{\epsilon}$
            $\hat{\mathbf{x}}_0 = (\mathbf{x}_1 - \sigma_1\boldsymbol{\epsilon}_{\boldsymbol{\theta}}(\mathbf{x}_1, \mathbf{c}, 1))/\alpha_1$
            $\boldsymbol{g} = \boldsymbol{g} + \nabla_{\boldsymbol{\theta}} r(\hat{\mathbf{x}}_0, \mathbf{c})$
    $\boldsymbol{\theta} \leftarrow \boldsymbol{\theta} - \eta\boldsymbol{g}$
**return** $\boldsymbol{\theta}$

---

**Gradient Checkpointing.** Gradient checkpointing (Gruslys et al., 2016; Chen et al., 2016) reduces the memory cost of storing activations for use during backprop at the cost of increased compute, by storing only a subset of activations in memory and re-computing the others on the fly. Specifically, we only store the input latent for each denoising step, and re-materialize the UNet activations during backprop. Implementing checkpointing in JAX (Bradbury et al., 2018) is as simple as adding `@jax.checkpoint` to the body of the sampling loop.

**DRaFT-$K$.** While gradient checkpointing makes it possible to backpropagate through the full sampling chain, we found that optimization speed and overall performance can be substantially improved by truncating backprop through only the last $K$ sampling steps. We call the resulting approach DRaFT-$K$, illustrated in Figure 2. Truncating backprop reduces the compute needed per step by decreasing the number of backward passes through the UNet. Surprisingly, it also improves training efficiency per-step, which we investigate further in Section 5.2. For small $K$ (e.g., $K = 1$), the memory cost of unrolling is small, and we do not need gradient checkpointing.

**DRaFT-LV.** Empirically, we found that simply setting $K = 1$ (i.e., only differentiating through the last sampling step) results in the best reward vs. compute tradeoff. Here, we propose a method for improving the efficiency of DRaFT-1 further by reducing the variance of the gradient estimate. We call this low-variance estimator DRaFT-LV. The key idea is to use the forward diffusion process to produce additional examples to train on without re-generating new images. Specifically, we noise the generated image $n$ times, and use the summed

reward gradient over these examples. Although DRaFT-LV adds $n$ additional forward and backward passes through the UNet and reward model, in practice this is fairly little overhead compared to the $T$ UNet calls already needed for sampling. We found that using $n = 2$ is around $2\times$ more efficient than DRaFT-1 while adding around 10% compute overhead for our reward functions.

**General Reward Fine-Tuning Algorithm.** Algorithm 1 presents a unified framework encompassing several gradient-based approaches for diffusion fine-tuning. In particular, it includes as special cases DRaFT, DRaFT-$K$, DRaFT-LV, and ReFL (Xu et al., 2023), which differ in how the gradient is computed. DRaFT-$K$ inserts a stop-gradient operation $\mathbf{x}_t = \texttt{stop\_grad}(\mathbf{x}_t)$ at sampling iteration $K$, to ensure that the gradient is only computed through the last $K$ steps (as the gradient does not flow through $\texttt{stop\_grad}$); vanilla DRaFT can be obtained by setting $K = T$. ReFL inserts a stop gradient and breaks out of the sampling loop early, returning a one-step predicted clean image. Additionally, it does so at a random timestep towards the end of sampling (we use $m = 20$ for 50-step sampling). In this unifying framework, ReFL with $m = 1$ is equivalent to DRaFT-1. Lastly, DRaFT-LV averages the last-step gradient over multiple examples with different noises to reduce variance. Note that in practice we use Adam and LoRA, not shown in the algorithm for simplicity.

## 5 EXPERIMENTS

In this section, we show that DRaFT can be applied to a wide range of reward functions, and that it substantially outperforms prior reward fine-tuning methods. We used Stable Diffusion 1.4 as the base diffusion model. As it is a latent diffusion model (Rombach et al., 2022), applying DRaFT involves backpropagating through both the sampling process that produces the final latent representation and the decoder used to obtain the image. DRaFT is applicable to any differentiable sampler; in our experiments, we used DDIM (Song et al., 2021a), but we found that ancestral sampling performs similarly. We used 50 sampling steps, so DRaFT-50 corresponds to backpropagating through the full sampling chain (i.e., vanilla DRaFT). We used classifier-free guidance weight 7.5. Experimental details, hyperparameters, and additional results are provided in Appendices A and B.

### 5.1 FINE-TUNING FOR AESTHETIC QUALITY

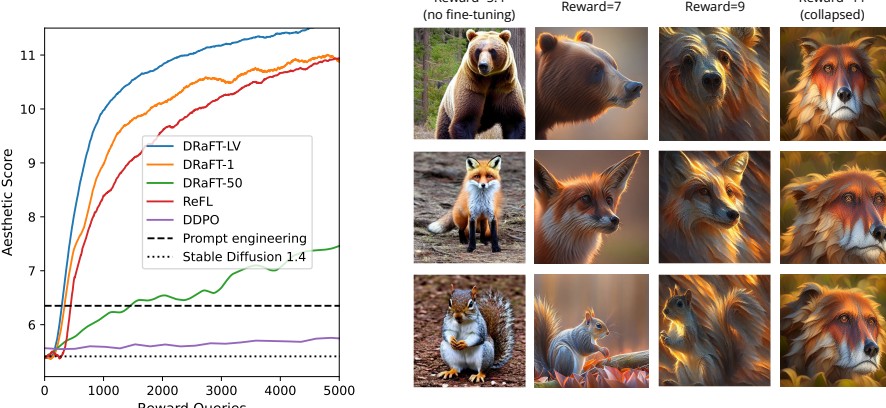

Figure 3: Left: Sample efficiency of methods when maximizing scores from the LAION Aesthetic Classifier. DDPO (Black et al., 2023) results were obtained from correspondence with the authors, as the results in the original paper use a buggy implementation; DDPO achieves a reward of 7.4 after 50k reward queries. Right: Qualitative comparison of generations as training progresses.

First, we compared fine-tuning methods to improve aesthetic quality scores given by the LAION aesthetic predictor, which is trained to rate images on a scale of 1 through 10. We used the simple set of 45 prompts from Black et al. (2023), where each prompt is the name of a common animal. We compared against the DDPO (Black et al., 2023) RL method, ReFL (Xu et al., 2023), and a prompt engineering baseline (see Appendix A.3 for more details). Results are shown in Figure 3. Because it does not make use of gradients, reinforcement learning is much less sample-efficient than DRaFT. Interestingly, truncating DRaFT to a single backwards step substantially improves sample efficiency; we analyze this phenomenon further in Section 5.2. Thanks to its lower-variance gradient estimate, DRaFT-LV further improves training efficiency. We observed that the methods initially produced improved images, but eventually collapsed to producing very similar high-reward images.

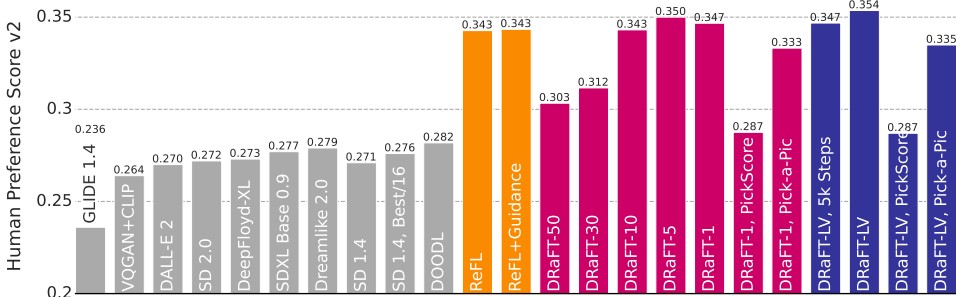

Figure 4: Comparison of the HPSv2 rewards achieved by a variety of baselines (denoted by gray bars), ReFL, DRaFT-$K$, and DRaFT-LV on the HPDv2 test set. All baselines except for DOODL and best of 16 are taken from Wu et al. (2023a). Unless indicated otherwise, models are Stable Diffusion 1.4 fine-tuned for 10k steps.

**Reward Hacking.** In Figure 3 (Right), we observe an instance of *reward hacking*, where the fine-tuned model loses diversity and collapses to generate a certain high-reward image. Reward hacking points to deficiencies in existing rewards; efficiently finding gaps between the desired behavior and the behavior implicitly captured by the reward is a crucial step in fixing the reward. Improved techniques for fine-tuning, such as DRaFT, can aid the development of new rewards. We discuss reward hacking further in Appendix B.8.

## 5.2 FINE-TUNING ON HUMAN PREFERENCES

Next, we applied DRaFT to two reward functions trained on human preference data: Human Preference Score v2 (HPSv2; Wu et al. 2023a) and PickScore (Beaumont et al., 2022).

**Qualitative comparison of reward functions.** First, we trained DRaFT models using the aesthetic, PickScore, and HPSv2 rewards on the same set of 45 animal prompts. As the reward models behave differently for photographs and paintings, we trained two versions of each model. Qualitatively, we found that the different reward functions led to quite different generated images (see Figure 5, Left). HPSv2 generally encourages more colorful but less photorealistic generations.

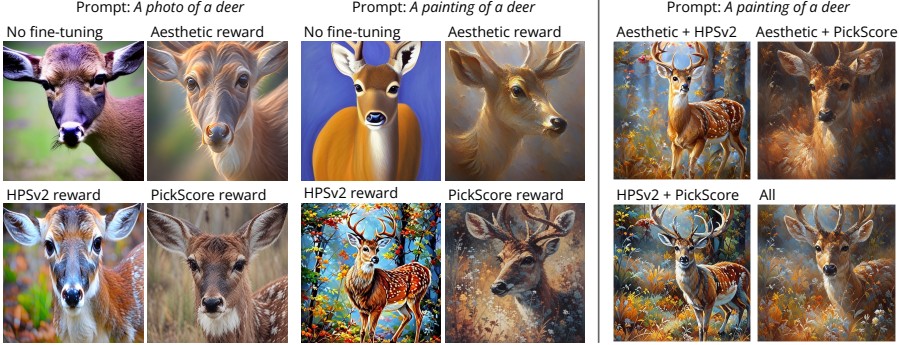

Figure 5: Left: Comparison of generations (using the same random seed) from DRaFT models fine-tuned using the LAION Aesthetic Classifier, Human Preference Score v2, and PickScore reward functions. Right: Outputs of DRaFT models combined by mixing the LoRA weights for different reward functions.

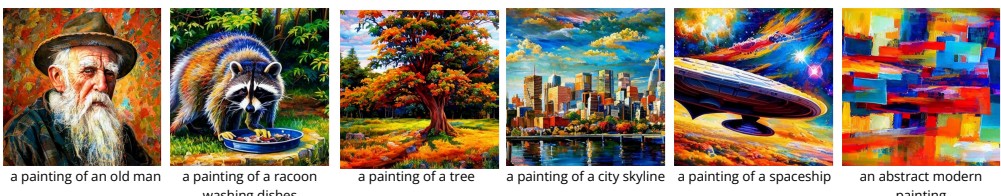

Figure 6: A DRaFT model trained on a simple prompt set of 45 animals generalizes well to other image subjects while preserving the learned style (Human Preference Score v2 reward).

**Human Preference Score Comparison.** Next, we compared DRaFT-finetuned models to several baselines on the Human Preference Score v2 benchmark (Wu et al., 2023a). We used DRaFT to

fine-tune for HPSv2, using the Human Preference Dataset v2 (HPDv2) traning set prompts. In Figure 4, we report the mean performance over the four test set categories; Table 2 in Appendix B.2 provides a detailed breakdown of results by category. We compared DRaFT to a selected set of baselines from Wu et al. (2023a), as well as to DOODL (Wallace et al., 2023) and ReFL (Xu et al., 2023). As ReFL does not use guidance during training, we also experimented with adding guidance to it. Interestingly, the model with guidance performs similarly; possibly fine-tuning compensates for the lack of guidance similarly to how guided diffusion models can be distilled into unguided ones (Meng et al., 2023). DRaFT-1 slightly outperforms ReFL, while being simpler to implement. DRaFT-LV achieves the best reward value, learning approximately $2\times$ faster than DRaFT-1 (see the "DRaFT-LV, 5k steps" bar). Figure 4 also provides an ablation comparing DRaFT-$K$ for $K \in \{1, 5, 10, 30, 50\}$. We found that using smaller $K$ was beneficial, both in terms of training time and final performance.

**Evaluating Generalization.** Next, we investigated several types of generalization: 1) *prompt generalization*, where we fine-tuned for HPSv2 on one prompt dataset (Pick-a-Pic) and evaluated performance on another prompt dataset (HPDv2)—these results are shown by the "DRaFT, Pick-a-Pic" bars in Figure 4; and 2) *reward generalization*, where we fine-tuned using a different reward (PickScore) than we used for evaluation (HPSv2)—these results are shown by the "DRaFT, PickScore" bars. DRaFT-1 and DRaFT-LV transfer well across prompt sets and fairly well across reward functions. We also show qualitative prompt generalization results in Figure 6: we found that a model fine-tuned using only 45 animal prompts learns a style that generalizes well to other prompts.

**Large-Scale Multi-Reward Model.** We pushed the scale of DRaFT-LV further by performing a $2\times$ longer training run on the larger HPDv2 dataset, optimizing for a weighted combination of rewards, PickScore $= 10$, HPSv2 $= 2$, Aesthetic $= 0.05$.[2] Images generated by the model are shown in Figure 1. Overall, the fine-tuned model produces more detailed and stylized images than baseline Stable Diffusion. In some cases, fine-tuning improves the model's image-text alignment as well as aesthetics, because the PickScore and HPSv2 rewards measure the similarity between the prompt and image embeddings. For example, the fine-tuned model almost always generates images faithful to the prompt "a raccoon washing dishes," while baseline Stable Diffusion does so less than half the time. However, we also found that the model sometimes overgeneralizes the reward function, generating detailed images even when the prompt calls for a simpler style (see Figure B.1 in Appendix B).

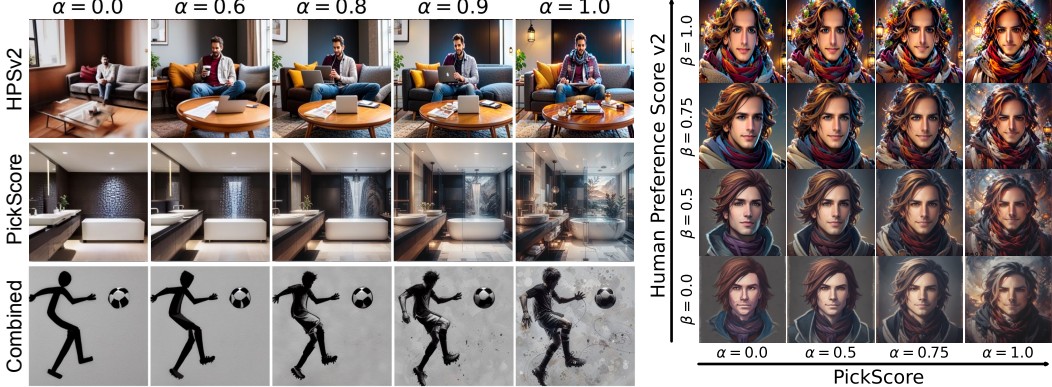

Figure 7: **Interpolating LoRA parameters.** Left: Generations from DRaFT models with different scalar multipliers applied to the LoRA weights. Scaling LoRA parameters yields semantic interpolations between the pre-trained and fine-tuned models. Right: Combining LoRA parameters that have been fine-tuned independently for two different rewards—PickScore and HPSv2—allows for fine-grained control over style without additional training. For a fixed noise sample, we show images generated using $\alpha\boldsymbol{\theta}_{\text{LoRA}}^{\text{PickScore}} + \beta\boldsymbol{\theta}_{\text{LoRA}}^{\text{HPSv2}}$.

**Scaling and Mixing LoRA Weights.** It is possible to control the strength of the fine-tuning simply by scaling the LoRA parameters: multiplying them by a scalar $\alpha < 1$ moves the adapted parameters closer to the original, pre-trained model. Figure 7 (Left) shows that this method smoothly interpolates between the pre-trained and DRaFT models. The LoRA scale could become a useful hyperparameter for controlling generations from DRaFT models similarly to how guidance strength is currently used.

---

[2]The gradients of the different rewards have substantially different norms, leading to the wide range of coefficients. The particular values were chosen based on sample quality after experimenting with a few settings.

While we experimented with KL regulariztaion and early stopping as alternative ways of reducing reward overfitting (see Figure 29 in Appendix B.8), we found LoRA scaling to work best.

We also probed the compositional properties of LoRA weights adapted for different rewards. One can combine the effects of multiple reward functions by interpolating the LoRA weights of the respective DRaFT models (Figure 5, Right). In Figure 7 (Right), we show images generated using linear combinations of LoRA parameters trained separately for the PickScore and HPSv2 rewards, using scaling coefficients $\alpha$ and $\beta$. The resulting images mix the semantics of HPSv2 and PickScore, and are simultaneously more saturated and detailed than the pre-trained model outputs. While it would also be possible to instead directly sum the score functions of the models (similar to Liu et al. 2022), this would increase generation time proportionally to the number of combined models.

**Understanding the Impact of $K$.** Interestingly, we found that truncated backpropagation improves sample efficiency as well as compute efficiency. Our analysis suggests that gradients explode as $K$ increases, which may lead to optimization difficulties for large $K$ (see Figure 8). We found that gradient clipping somewhat alleviates this issue (see Figure 28 in Appendix B.7), but does not fully close the gap. In Figure 9, we show an ablation over $K$ for aesthetic fine-tuning: performance degrades as $K$ increases for $K > 10$. In Figure 10, we show that although DRaFT-1 only computes gradients through the final generation step, it generalizes across time and affects the whole sampling chain. We compare samples generated with different

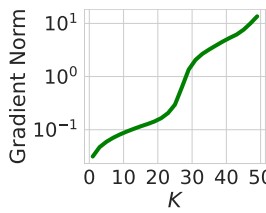

Figure 8: Gradient norms

*LoRA start iterations $M$*, where we applied the LoRA-adapted diffusion parameters for the last $M$ steps of the sampling chain, while in the first $T - M$ steps we used the pre-trained Stable Diffusion parameters without LoRA adaptation. We observe that adaptation does not happen only at the end of sampling (also see Figure 26 in App. B.7). We also investigated the opposite scenario, where the LoRA-adapted parameters are used in the first $M$ steps, after which LoRA is "turned off" and we use only the original pre-trained parameters for the remaining $T - M$ steps (see Figure 27 App. B.7), which shows that it is also important to apply the LoRA parameters early in sampling.

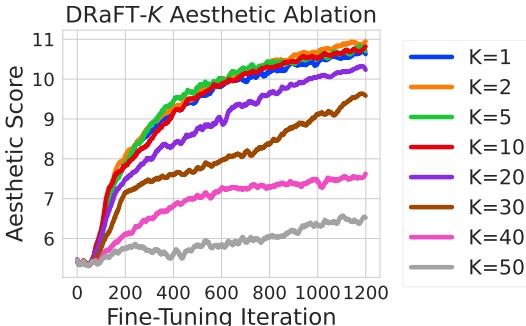

Figure 9: Ablation over $K$ for aesthetic reward fine-tuning.

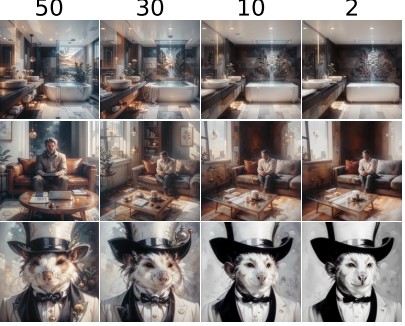

Figure 10: Images generated with different LoRA start steps $M \in \{50, 30, 10, 2\}$.

### 5.3 OTHER REWARD FUNCTIONS

**Compressibility and Incompressibility.** Inspired by Black et al. (2023), we also fine-tuned a diffusion model to generate easily-compressible images. While the JPEG compression and decompression algorithms available in common libraries are not differentiable by default, they can be implemented in a differentiable manner using deep learning libraries like PyTorch (Paszke et al., 2019) or JAX (Bradbury et al., 2018). Given a prompt $c$, we pass the output of the diffusion model, $x_0 = \text{sample}(\theta, c, x_T)$ through differentiable compression and decompression al-

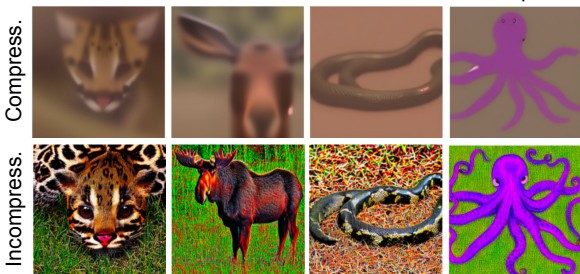

Figure 11: Top: Images generated by a diffusion model fine-tuned for JPEG compressibility, with reward $r(x_0) = -\|x_0 - d(c(x_0))\|_2^2$ where $x_0 = \text{sample}(\theta, c, x_T)$. Bottom: Using the *negation* of the compressibility loss yields images that are difficult to compress.

gorithms to obtain $d(c(\mathbf{x}_0))$, and minimize the Euclidean distance between the original and reconstructed images. This encourages the diffusion model to produce simple images. The results using DRaFT to fine-tune for JPEG compressibility are shown in Figure 11 (Top): these images have simple, uniform-color backgrounds and low-frequency foregrounds. In addition, negating the compressibility loss yields *incompressible* images, shown in Figure 11 (Bottom), that have complex, high-frequency foregrounds and backgrounds. Additional details are provided in Appendix B.3.

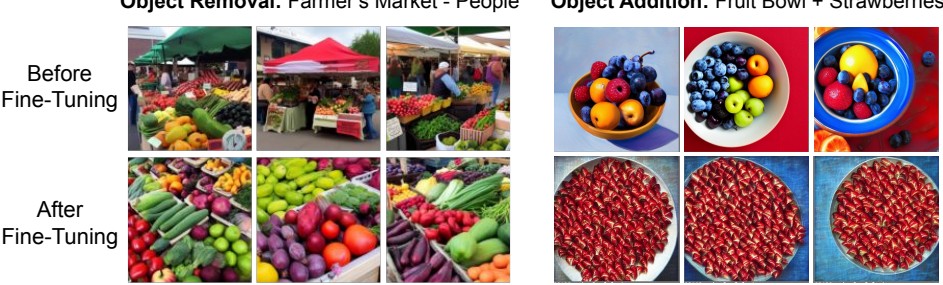

**Object Removal:** Farmer's Market - People     **Object Addition:** Fruit Bowl + Strawberries

Before Fine-Tuning

After Fine-Tuning

Figure 12: **Fine-tuning for object removal or addition using the OWL-ViT object detector.** Left: The prompt to the diffusion model is `a farmer market` and the queries for OWL-ViT are $\mathcal{Q} = \{$`a person`, `people`$\}$. Before fine-tuning, most of the generated images contain people in the background; after finetuning, all images contain only vegetables. Right: Adding strawberries to a fruit bowl.

**Object Detection and Removal.** Here, we explore the use of a pre-trained object detection model to bias a diffusion model's generations to include or exclude a certain object class. We use OWL-ViT (Minderer et al., 2022), an open-world localization model that takes arbitrary text queries and returns bounding boxes with scores for the corresponding objects in the image. We pass images generated by the diffusion model through a pre-trained OWL-ViT model, together with a set of queries $\mathcal{Q}$ that we wish to exclude from the generated images (see Figure 22 in Appendix B.6 for details). As the reward, we use the sum of scores for the localized objects corresponding to all queries $q \in \mathcal{Q}$, as well as the areas of their bounding boxes. In Figure 12 (Left), we show the results of fine-tuning to minimize detection of people. Similarly, we can maximize the sum of scores for localized objects to encourage generations to *include* a certain class: in Figure 12 (Right), we use the prompt "a bowl of fruit" and queries $\mathcal{Q} = \{$strawberry$\}$.

**Adversarial Examples.** Our framework also provides an avenue to study the inductive biases of both diffusion models and pre-trained classifiers: we can fine-tune a diffusion model such that images generated based on a prompt for a class $y$ (e.g., "mouse") are classified as a different class $y'$ (e.g., "cat") by a ResNet-50 pretrained on ImageNet. As the fine-tuning reward, we use the negative cross-entropy to the target class. From the qualitative results shown in Figure 13, we observe that this classifier is texture-biased (Geirhos et al., 2019), as the images generated by the fine-tuned model incorporate cat-like textures while keeping the animal shapes mostly unchanged.

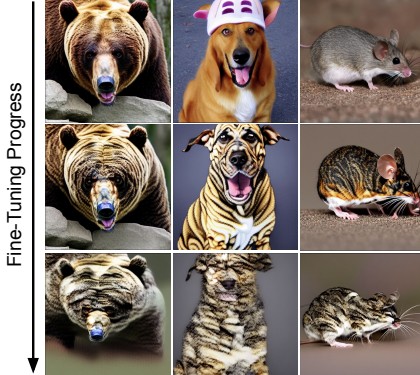

Fine-Tuning Progress

Figure 13: Diffusion adversarial examples for prompts `bear`, `dog`, `mouse` with target class `cat`.

## 6 CONCLUSION

We introduced an efficient framework for fine-tuning diffusion models on differentiable rewards by leveraging reward gradients. Our first method, Direct Reward Fine-Tuning (DRaFT), backpropagates through the full sampling procedure, and achieves strong performance on a variety of reward functions. We then propose DRaFT-$K$ and DRaFT-LV, variants which improve efficiency via truncated backpropagation through time. In addition, we draw connections between our approach and prior work such as ReFL (Xu et al., 2023), to provide a unifying perspective on the design space of gradient-based reward fine-tuning algorithms. We show that various methods can be obtained as special cases of a more general algorithm, depending on the whether—and where—`stop_gradient` operations are included. We hope that our work will inspire the development of improved techniques for reward fine-tuning; just as RLHF has become crucial for deploying large language models, we believe that reward fine-tuning may become a key step for improving image generation models.

ACKNOWLEDGEMENTS

We thank Daniel Watson, Jason Baldridge, Hartwig Adam, Robert Geirhos, and the anonymous reviewers for their thoughtful comments and suggestions. We thank Xiaoshi Wu for help with the HPSv2 dataset, Jiazheng Xu for answering our questions about ReFL, and Kevin Black for answering our questions about DDPO and re-running their aesthetic reward experiments.

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

APPENDIX

This appendix is structured as follows:

- In Appendix A, we provide more details of our experimental setup, including hyperparameters and baselines.

- In Appendix B, we provide additional results and analysis.

- In Appendix C, we discuss two other methods we explored that did not achieve as strong results as DRaFT-LV: a single-reward version of DRaFT-LV and applying Deterministic Policy Gradient.

- In Appendix D, we provide an extended discussion of related work.

- In Appendix E, we provide uncurated samples from models fine-tuned for various reward functions, including HPSv2, PickScore, and a combined reward weighting HPSv2, PickScore, and Aesthetic score.

## A EXPERIMENTAL SETUP

### A.1 ADDITIONAL EXPERIMENTAL DETAILS AND HYPERPARAMETERS

Here we expand on training details and provide hyperparameters. We used the same hyperparameters for ReFL and our DRaFT variants.

**Optimization.** We use the AdamW optimizer (Loshchilov & Hutter, 2019) with $\beta_1 = 0.9$, $\beta_2 = 0.999$, and a weight decay of 0.1. As we use LoRA, the weight decay regularizes fine-tuning by pushing parameters towards the pre-trained weights. For the longer training runs (10k steps on human preference reward functions), we use a square root learning rate decay, scaling the learning rate by $\min\left(10 \cdot \text{step}^{-0.5}, 1\right)$. Weight decay and learning rate decay slightly improve results, but are not critical to the method. During training, we convert the pre-trained Stable Diffusion parameters to bfloat16 to reduce the memory footprint, but use float32 for the LoRA parameters being trained.

**LoRA.** We apply LoRA to the feedforward layers as well as the cross-attention layers in the UNet, which we found to produce slightly better results. We apply LoRA only to the UNet parameters, not to the CLIP text encoder or VAE decoder mapping latents to images, as these did not improve results in initial experiments.

**Sampling.** We use 50 steps of DDIM sampling (where no noise is added during sampling beyond the initial sampled noise). We found the results using ancestral/DDPM sampling to be similar. We employ classifier-free guidance with a guidance weight of 7.5 both at train-time and test-time.

**Hyperparameters.** We apply DRaFT in two settings: large-scale (human preference reward functions using the HPSv2 or PickScore prompt sets) and small-scale (the other experiments). Hyperparameters are listed in Table 1. Small-scale training runs take around 1.5 hours on 4 TPUv4s. Large-scale training runs take around 8 hours on 16 TPUv4s.

| Hyperparameter | Small-Scale Experiments | Large-scale Experiments |
|---|:---:|:---:|
| Learning rate | 4e-4 | 2e-4 |
| Batch size | 4 | 16 |
| Train steps | 2k | 10k |
| LoRA inner dimension | 8 | 32 |
| Weight decay | 0.1 | 0.1 |
| DDIM steps | 50 | 50 |
| Guidance weight | 7.5 | 7.5 |
| DRaFT-LV inner loops $n$ | 2 | 2 |
| ReFL max timestep $m$ | 20 | 20 |

Table 1: Training hyperparameters.

### A.2 REWARD FUNCTION DETAILS

The **LAION aesthetic predictor** is trained to rate images on a scale of 1 through 10. The predictor consists of a feedforward neural network on top of a CLIP (Radford et al., 2021) image encoder pre-trained on a variety of human-labeled sources.

**Human Preference Score v2** (HPSv2; Wu et al. 2023a) is trained on prompts from DiffusionDB (Wang et al., 2023b) and COCO Captions (Chen et al., 2015). Images are generated from each using a wide range of generative models. Human annotators then rate which of two images is preferred for a prompt. The reward model is OpenCLIP-H (Beaumont et al., 2022) fine-tuned on the preference judgements. Unlike the aesthetic reward, HPSv2 takes into account the prompt as well as the image.

**PickScore** (Kirstain et al., 2023) also uses OpenCLIP-H fine-tuned on preference data. However, the data is collected in a more organic way through a web app, where users can generate and then rate pairs of images generated from different diffusion models with different sampler hyperparameters.

### A.3 BASELINES

**Denoising Diffusion Policy Optimization (DDPO; Black et al. 2023)** essentially applies the RE-INFORCE policy gradient algorithm (Williams, 1992) to diffusion model generation. The method trains the model to increase the likelihoods of actions (i.e. sampling steps) that produce high-reward samples. DDPO employs additional tricks such as importance weight clipping, reward normalization, and classifier-free guidance-aware training to improve results. We show results from DDPO-IS, which extends DDPO to use an importance sampling estimator for learning on examples generated from a stale policy, enabling multiple optimization updates per reward query. Empirically, DDPO was found to outperform previous supervised approaches like reward weighted regression (Lee et al., 2023). Rather than using the results in Figure 4 of the original paper, we obtained results from direct correspondence with the authors, as their original aesthetic reward implementation had a bug.[3]

**Prompt Engineering.** Users often improve the aesthetic quality of generated images through prompt engineering. We perform semi-automated prompt search with access to the reward function, trying out various prompts and selecting the one with the best aesthetic score. The best-scoring prompt we found is `A stunning beautiful oil painting of a {:}, cinematic lighting, golden hour light.`

**Re-ranking (Best of 16).** This simple baseline generates 16 images for each caption with the pre-trained Stable Diffusion model and selects the one with the highest reward.

**Direct Optimization of Diffusion Latents (DOODL).** DOODL (Wallace et al., 2023) backpropagates through diffusion sampling to update the initial noise $\mathbf{x}_T$ so that the resulting generation produces a high reward. Rather than using an invertible diffusion algorithm to reduce memory costs as in Wallace et al. (2023), we simply use gradient checkpointing as with DRaFT. We optimize $\mathbf{x}_T$ with Adam (which worked slightly better for us than momentum only) over 20 steps, which makes generation approximately $60\times$ slower than standard sampling. Following Wallace et al. (2023), we renormalize the latent at every step after the Adam update.

## B ADDITIONAL EXPERIMENTAL RESULTS

### B.1 OVER-GENERALIZATION

Here we show examples of overgeneralization, where fine-tuning on preferences trains the model to generate detailed images, even when the prompt asks for a simple drawing.

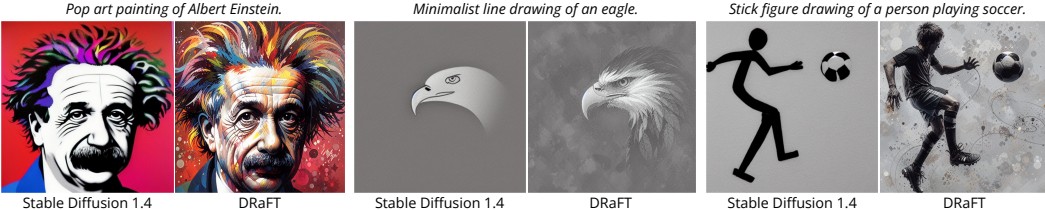

*Pop art painting of Albert Einstein.*     *Minimalist line drawing of an eagle.*     *Stick figure drawing of a person playing soccer.*

Stable Diffusion 1.4     DRaFT          Stable Diffusion 1.4     DRaFT          Stable Diffusion 1.4     DRaFT

---

[3]See https://github.com/kvablack/ddpo-pytorch/issues/3#issuecomment-1634723127

## B.2 Full Quantitative Results

In Table 2, we provide quantitative results comparing DRaFT to several baselines on the Human Preference Score v2 benchmark from Wu et al. (2023a).

| Model | Animation | Concept Art | Painting | Photo | Averaged |
|---|---|---|---|---|---|
| GLIDE 1.4 | 0.2334 | 0.2308 | 0.2327 | 0.2450 | 0.2355 |
| VQGAN + CLIP | 0.2644 | 0.2653 | 0.2647 | 0.2612 | 0.2639 |
| DALL-E 2 | 0.2734 | 0.2654 | 0.2668 | 0.2724 | 0.2695 |
| SD 2.0 | 0.2748 | 0.2689 | 0.2686 | 0.2746 | 0.2717 |
| DeepFloyd-XL | 0.2764 | 0.2683 | 0.2686 | 0.2775 | 0.2727 |
| SDXL Base 0.9 | 0.2842 | 0.2763 | 0.2760 | 0.2729 | 0.2773 |
| Dreamlike Photoreal 2.0 | 0.2824 | 0.2760 | 0.2759 | 0.2799 | 0.2786 |
| Stable Diffusion (SD 1.4) | 0.2726 | 0.2661 | 0.2666 | 0.2727 | 0.2695 |
| ReFL | 0.3446 | 0.3442 | 0.3447 | 0.3374 | 0.3427 |
| DRaFT-50 | 0.3078 | 0.3021 | 0.3019 | 0.3013 | 0.3033 |
| DRaFT-30 | 0.3148 | 0.3112 | 0.3109 | 0.3096 | 0.3116 |
| DRaFT-10 | 0.3456 | 0.3447 | 0.3452 | 0.3369 | 0.3431 |
| DRaFT-5 | 0.3521 | 0.3520 | 0.3526 | 0.3431 | 0.3500 |
| DRaFT-1 | 0.3494 | 0.3488 | 0.3485 | 0.3400 | 0.3467 |
| DRaFT-1, PickScore | 0.2907 | 0.2848 | 0.2861 | 0.2882 | 0.2875 |
| DRaFT-1, Pick-a-Pic | 0.3399 | 0.3358 | 0.3385 | 0.3188 | 0.3332 |
| DRaFT-LV | 0.3551 | 0.3552 | 0.3560 | 0.3480 | 0.3536 |
| DRaFT-LV, PickScore | 0.2880 | 0.2833 | 0.2846 | 0.2923 | 0.2870 |
| DRaFT-LV, Pick-a-Pic | 0.3409 | 0.3371 | 0.3408 | 0.3210 | 0.3349 |

Table 2: Following Wu et al. (2023a), we used the HPDv2 training set to fine-tune our models, and we evaluated performance on four benchmark datasets: Animation, Concept Art, Paintings, and Photos. We report the average score achieved for each benchmark, as well as the aggregated performance across all datasets. The results above the separator are taken from Wu et al. (2023a); the results below were obtained from our implementations and experiments.

## B.3 Compressibility and Incompressibility Details

Figure 14 illustrates the setup we used to fine-tune for JPEG compressibility and incompressibility. Given a prompt $\mathbf{c}$, we take the image generated by the diffusion model, $\mathbf{x}_0 = \text{sample}(\boldsymbol{\theta}, \mathbf{c}, \mathbf{x}_T)$ and pass it through JPEG compression and decompression functions, denoted by $c$ and $d$, respectively. As the reward, we use the negative Euclidean distance between the original sample $\mathbf{x}_0$ and its reconstruction $r(\mathbf{x}_0) = -\|\mathbf{x}_0 - d(c(\mathbf{x}_0))\|^2$. While the JPEG implementations found in common libraries are not directly differentiable, the JPEG algorithm can be re-implemented in a differentiable way using autodiff frameworks; we re-implemented JPEG compression and decompression in JAX (Bradbury et al., 2018). Figures 15 and 16 show additional samples generated by diffusion models fine-tuned for compressibility and incompressibility, respectively, and Figure 17 shows the results of overoptimization for incompressibility. In our experiments, we used a fixed compression ratio of $4\times$.

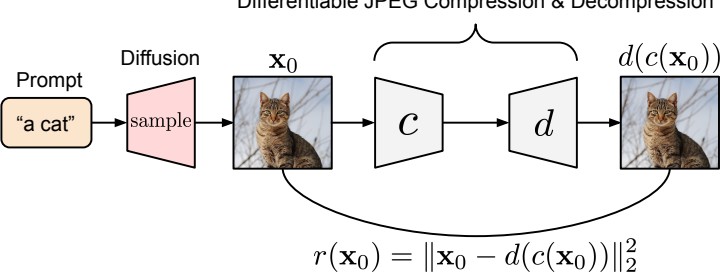

Figure 14: Fine-tuning for JPEG compressibility, using differentiable JPEG compression and decompression algorithms.

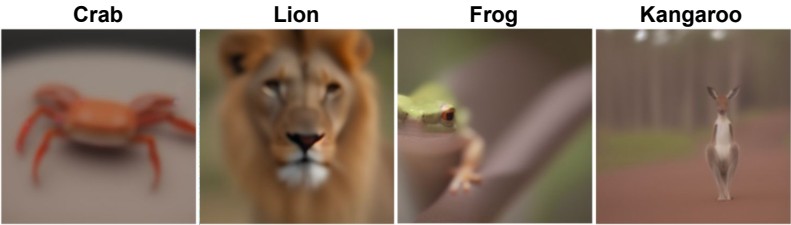

Figure 15: Images generated by a diffusion model fine-tuned for JPEG compressibility.

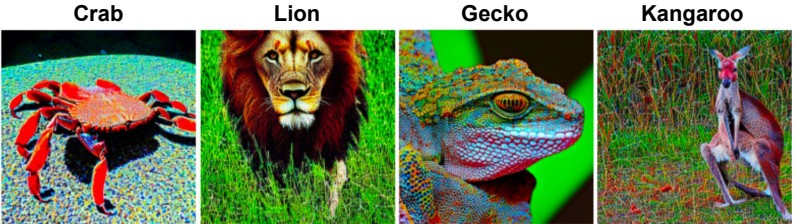

Figure 16: Images generated by a diffusion model fine-tuned using the *negation* of the JPEG compressibility reward, which yields images that are *difficult to compress*. We observe high-frequency foreground and background information.

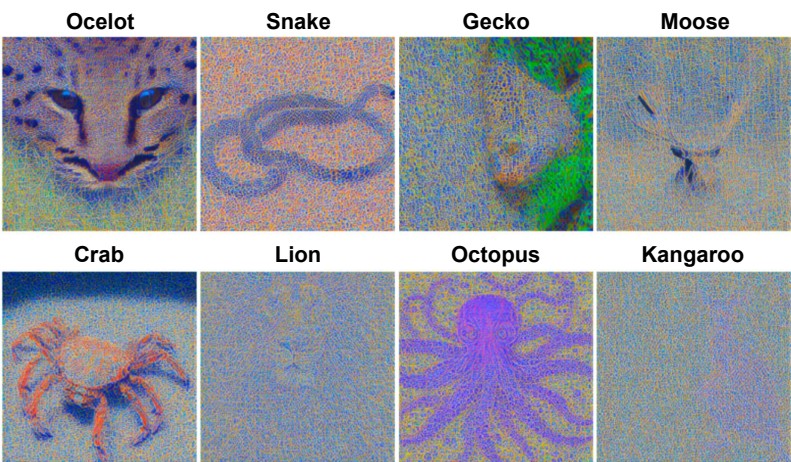

Figure 17: Over-optimization for JPEG incompressibility can lead to images with strong high-frequency texture and a loss of semantic information. Interestingly, here the octopus is generated with more than eight arms.

## B.4 ADVERSARIAL EXAMPLE DETAILS

The setup for our diffusion adversarial example experiments is illustrated in Figure 18. We used a ResNet-50 model pre-trained on ImageNet-1k to classify images generated by Stable Diffusion; as the reward for fine-tuning, we used the negative cross-entropy to a fixed target class.

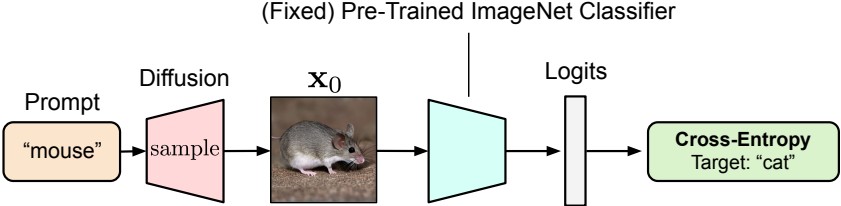

Figure 18: Generating "diffusion adversarial examples" that are conditioned on a text prompt (e.g., "mouse") but must be classified by a pre-trained ImageNet classifier as a different class (e.g., "cat").

## B.5 LoRA Scaling and Interpolation

Here, we provide additional results for scaling and interpolating LoRA parameters. Figures 19 and 20 show the effect of scaling LoRA parameters to control the strength of reward adaptation for the Human Preference Score v2 and PickScore rewards, respectively, and Figure 21 shows additional interpolations using linear combinations of LoRA weights trained separately for each of these rewards.

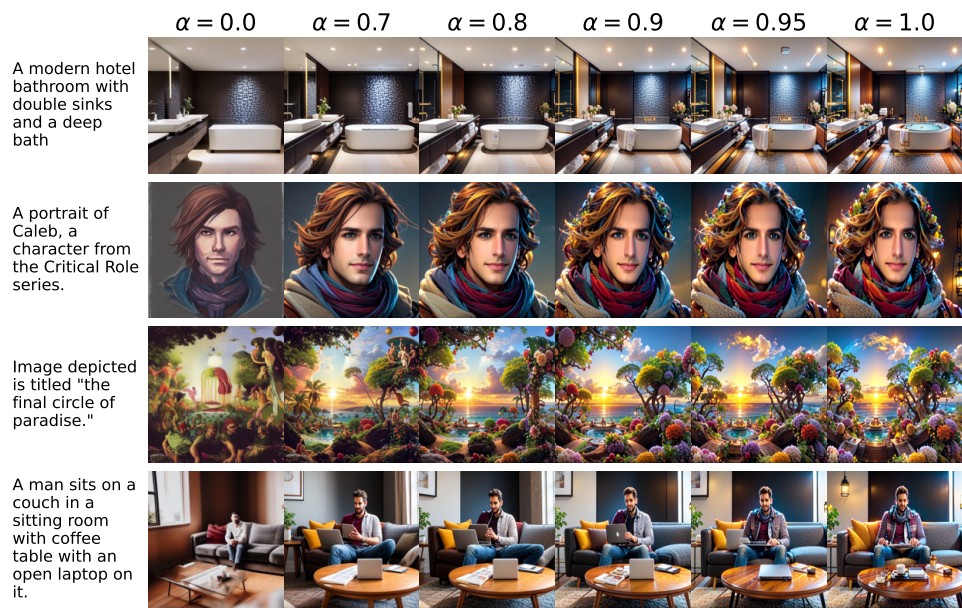

Figure 19: Scaling LoRA parameters (here adapted for Human Preference Score v2 (Wu et al., 2023a)) yields semantic interpolations between the original Stable Diffusion model and the fine-tuned model.

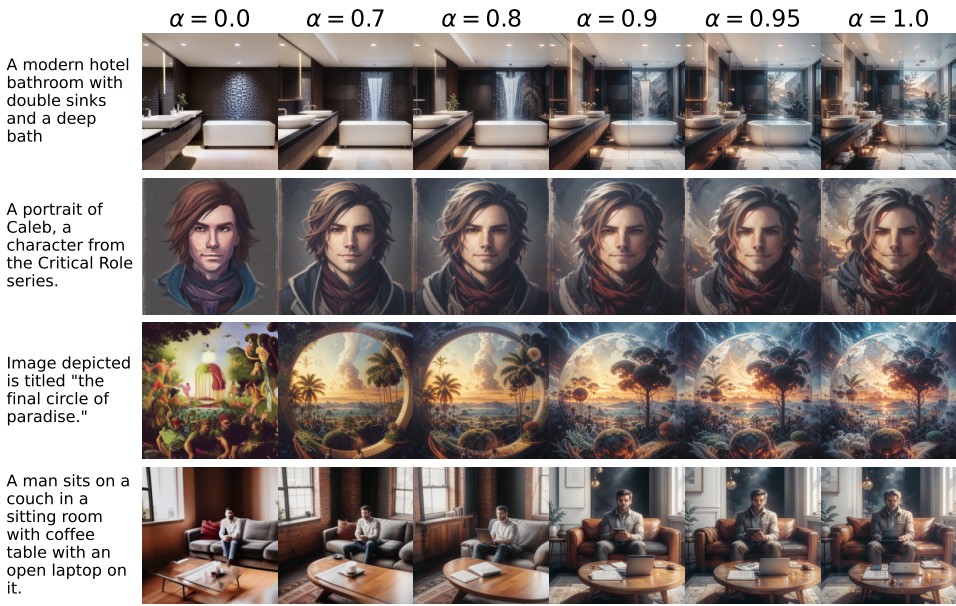

Figure 20: Scaling LoRA parameters (here adapted for PickScore (Kirstain et al., 2023)) yields semantic interpolations between the original Stable Diffusion model and the fine-tuned model.

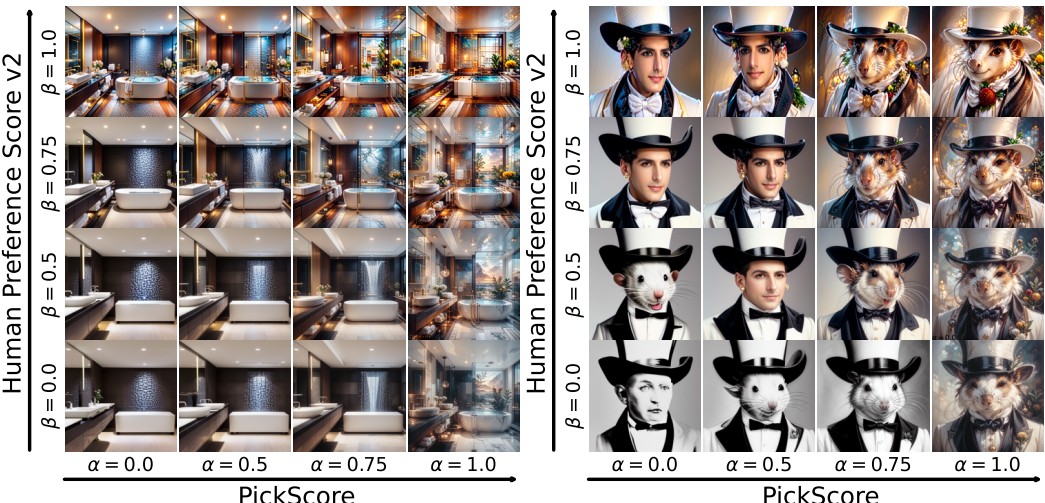

Figure 21: Additional results using linear combinations of LoRA parameters adapted for different rewards. In each subplot, we show images generated using LoRA parameters $\alpha\boldsymbol{\theta}_{\text{LoRA}}^{\text{PickScore}} + \beta\boldsymbol{\theta}_{\text{LoRA}}^{\text{HPSv2}}$ for coefficients $\alpha, \beta \in \{0.0, 0.5, 0.75, 1.0\}$. We found that interpolating between LoRA weights can yield smooth transitions between different styles.

## B.6 OBJECT DETECTION

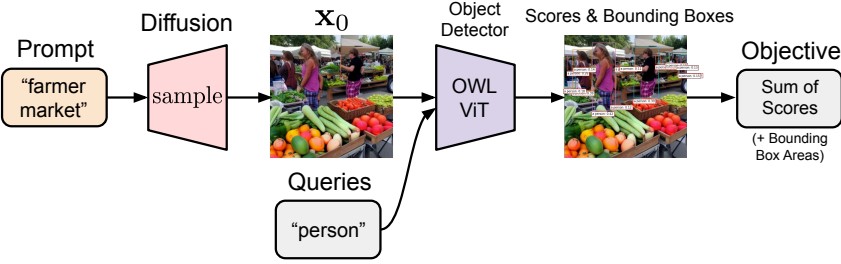

Figure 22: Fine-tuning for object detection using a pre-trained OWL-ViT model (Minderer et al., 2022).

Our setup for object detection is illustrated in Figure 22. We feed the generated images into a pre-trained Open-World Localization Vision Transformer (OWL-ViT) (Minderer et al., 2022) model, along with a list of text queries that we wish to localize in the image. OWL-ViT outputs bounding boxes and scores for the localized objects; we experimented with reward functions based on the sum of scores, as well as based on the areas of the bounding boxes, and found that both worked well. Figure 23 shows the fine-tuning progress for the diffusion prompt "a bowl of fruit" with OWL-ViT query "strawberries."

## B.7 UNDERSTANDING THE IMPACT OF $K$

We investigated the impact of $K$ on the behavior of models fine-tuned with DRaFT-$K$ via ablations designed to answer the following questions: 1) Does adaptation occur only in the last $K$ steps (e.g., does the initial segment of

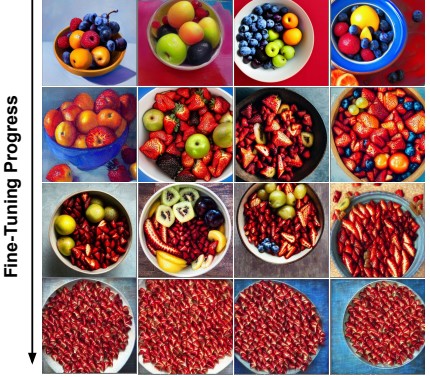

Figure 23: Maximizing the object detector score for the query "strawberry" yields fruit bowls that contain progressively more strawberries over the course of fine-tuning.

the sampling chain behave similarly to the original pre-trained model, with a sudden jump occurring at the end)? and 2) Is it necessary to apply the LoRA parameters throughout the full sampling chain? What is the impact of applying the LoRA parameters to only the initial or final portions of the trajectory?

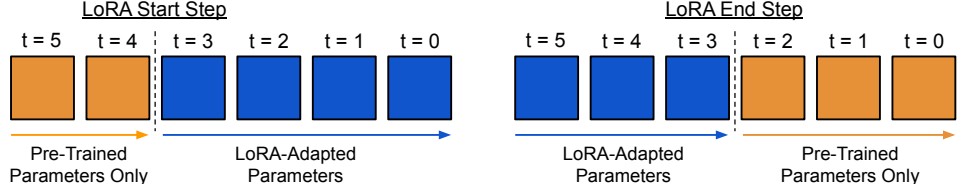

Figure 24: Illustration of sampling with different LoRA start steps and end steps.

To answer these questions, we applied the LoRA-adapted diffusion parameters for the last $M$ steps of the sampling chain, while in the first $T - M$ steps we used the pre-trained Stable Diffusion parameters without LoRA adaptation (see Figure 24). Figure 26 compares samples generated with different *LoRA start iterations* $M$; interestingly, although the LoRA parameters are fine-tuned using truncated backpropagation through only the last $K$ steps of sampling, adaptation does *not* happen only in the last $K$ steps; rather, the LoRA parameters need to be applied for at least 10-20 steps to yield substantial changes in the generated images. Similarly, we also investigated the opposite scenario, where the LoRA-adapted parameters are only applied in the first $M$ steps of the sampling chain, after which LoRA is "turned off" and we use only the original pre-trained parameters for the remaining $T - M$ steps. The generated images for a range of *end LoRA steps* $M \in \{2, 5, 10, 20, 30, 40, 50\}$ are shown in Figure 27. These results further demonstrate the importance of applying LoRA parameters in the early part of the sampling chain.

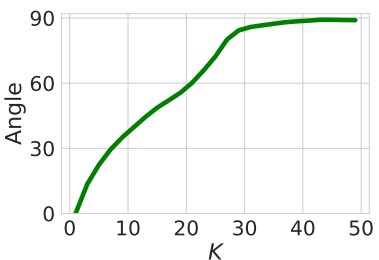

Figure 25: Angles between the DRaFT-1 gradient and DRaFT-$K$ gradients for $K \in \{1, \ldots, 50\}$.

In Figure 25, we show the angle between the DRaFT-1 gradient and DRaFT-$K$ gradients for various $K$; the gradients become nearly orthogonal for $K > 30$.

**Generated Images**

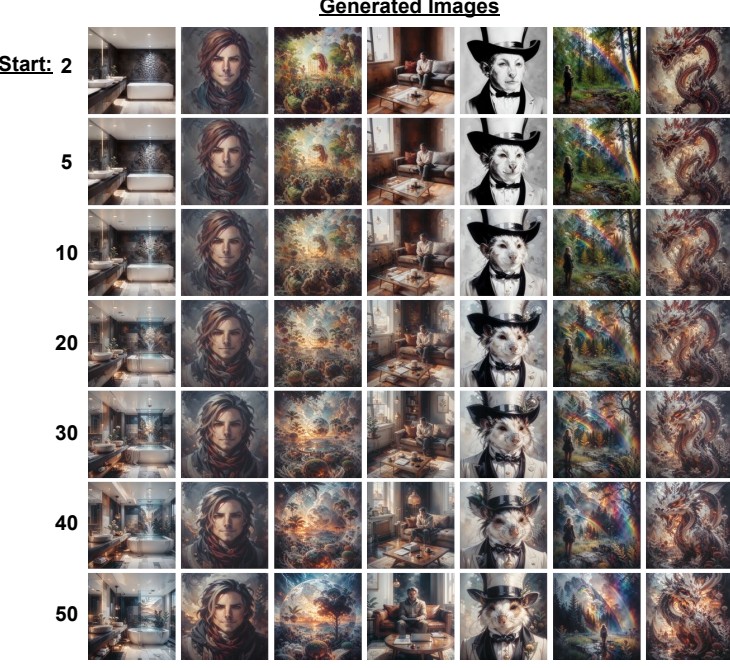

Figure 26: **Images generated with different LoRA start steps.** Here, the first $T - M$ steps of the diffusion sampling chain use the pre-trained diffusion model parameters, while the last $M$ steps use the LoRA-adapted parameters. The LoRA start step $M$ increases from top to bottom, $M \in \{2, 5, 10, 20, 30, 40, 50\}$. We observe that, although the LoRA parameters are trained via truncated backpropagation through the last few steps of sampling, they affect the entire sampling chain—the images improve qualitatively when the LoRA-adapted parameters are used for many steps (e.g., $M > 10$).

**Generated Images**

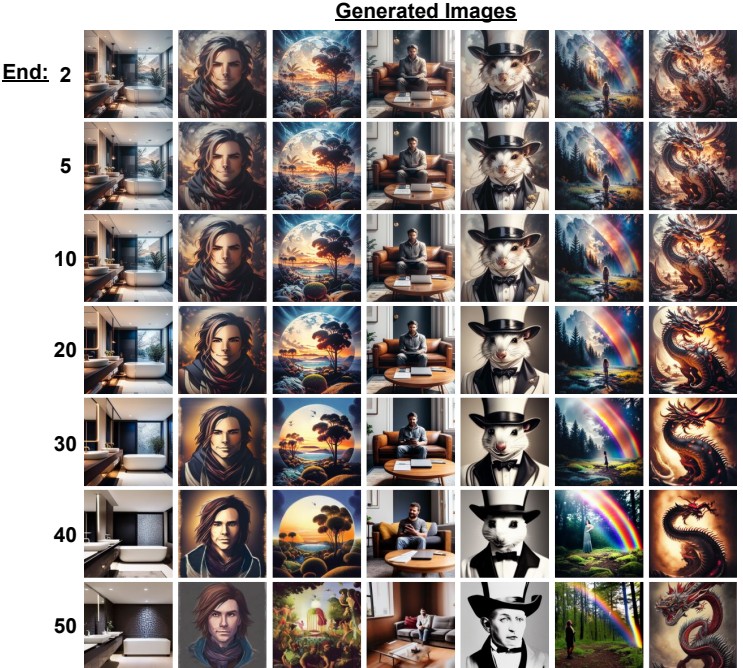

Figure 27: **Images generated with different LoRA end steps.** Here, the first $M$ steps of the diffusion sampling chain use the LoRA-adapted diffusion parameters, while the last $T - M$ steps use the un-adapted pretrained model parameters. The LoRA end step $M$ increases from top to bottom, $M \in \{2, 5, 10, 20, 30, 40, 50\}$. Interestingly, we observe that it is beneficial to apply the LoRA parameters early in sampling.

**Gradient Clipping.** We used gradient clipping throughout all of our experiments. In Figure 28, we show an ablation over the gradient clipping norm hyperparameter, which we denote $c$. We compared the reward values achieved over the course of fine-tuning for PickScore on the HPD-v2 prompt dataset, with gradient clipping norms $c \in \{0.001, 0.01, 0.1, 1.0, 10.0, 100.0, 1000.0\}$. We found that, when using DRaFT-50, smaller gradient clipping norms ($c = 0.001$) improved optimization substantially, while large clipping norms ($c = 1000$) impeded training. This further supports the observation that backpropagating through long diffusion sampling chains can lead to exploding gradients. We performed a similar ablation for DRaFT-1, and found that the fine-tuning performance was nearly identical for all gradient clipping coefficients ($c = 0.001$ to $c = 1000.0$).

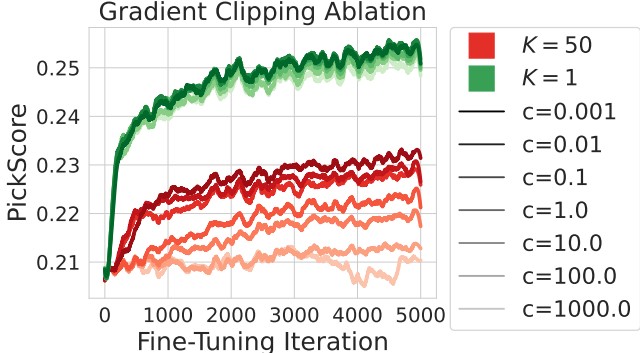

Figure 28: Ablation over the gradient clipping norm $c$ using DRaFT-$K$, fine-tuning to maximize PickScore on the HPD-v2 prompt dataset. DRaFT-50 experiments are shown in red, and DRaFT-1 experiments are shown in green. In both cases, darker colors denote smaller gradient clipping norms.

## B.8 "Over-Optimization" and Reward Hacking

**Improving Cross-Reward Generalization.** DRaFT can cause models to overfit to the reward function, leading to high-reward but subjectively lower-quality images. Here, we explore regularization methods to reduce reward overfitting.

As an approximate quantitative measure for over-optimization, we trained models on one reward function (HPSv2) and evaluated their generalization to another reward (PickScore). We considered three strategies to mitigate reward overfitting:

- **LoRA Scaling**: multiplying the LoRA weights by a constant $< 1$ (see Section 5.2), which interpolates between the pre-trained model and fine-tuned model, similarly to Wortsman et al. (2022).
- **Early stopping**: evaluating various checkpoints across the training run instead of only the last one; early ones will achieve lower train reward but hopefully generalize better.
- **KL Regularization**: similarly to Fan et al. (2023), we add a loss term to the model that penalizes the KL divergence between the pre-trained and fine-tuned model distributions. This amounts to a squared error term between the pre-trained and fine-tuned predicted $\epsilon$. We trained several separate models, each with a different weight on the KL loss, which is a disadvantage over the previous methods that can be applied to a single training run.

All models used DRaFT-LV with the large-scale hyperparameters (see Table 1). The results are shown in Figure 29. Both early stopping and KL regularization are commonly used to reduce reward overfitting in large language model preference tuning, but we found LoRA scaling to be more effective than the other approaches.

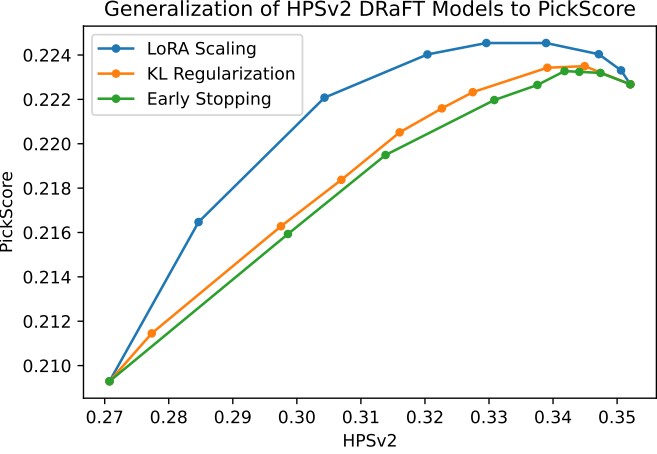

Figure 29: Generalization from HPSv2 to PickScore for various regularization methods used to prevent reward overfitting.

**Attempts to Increase Diversity.** A common result of reward hacking is a collapse to a single, high-reward image. Here, we describe two of our attempts to address this lack of diversity. First, we tried adding dropout to the reward function (in this case, the aesthetic classifier); however, we found that even large dropout rates such as 0.95 led to diversity collapse, as shown in Figure 30 (Left). Next, we tried adding a term to the reward that measures the dissimilarity between images generated in the same minibatch—this encourages the model to generate different images for different prompts and noise samples. Ideally, one would take the mean dissimilarity over all pairs of examples in a minibatch, yielding $\binom{B}{2}$ combinations where $B$ is the minibatch size. However, for computational tractability, we used a simpler approach, where we formed pairs by reversing the elements of a minibatch and computing elementwise dissimilarities, giving us $B/2$ pairs. We tried two measures of dissimilarity: 1) the Euclidean distance, and 2) the Learned Perceptual Image Patch Similarity (LPIPS) (Zhang et al., 2018). Neither was satisfactory to increase diversity: in Figure 30 (Right), we

show images generated using a large weight on the LPIPS diversity term, which increased inner-batch diversity at the expense of the aesthetic score.

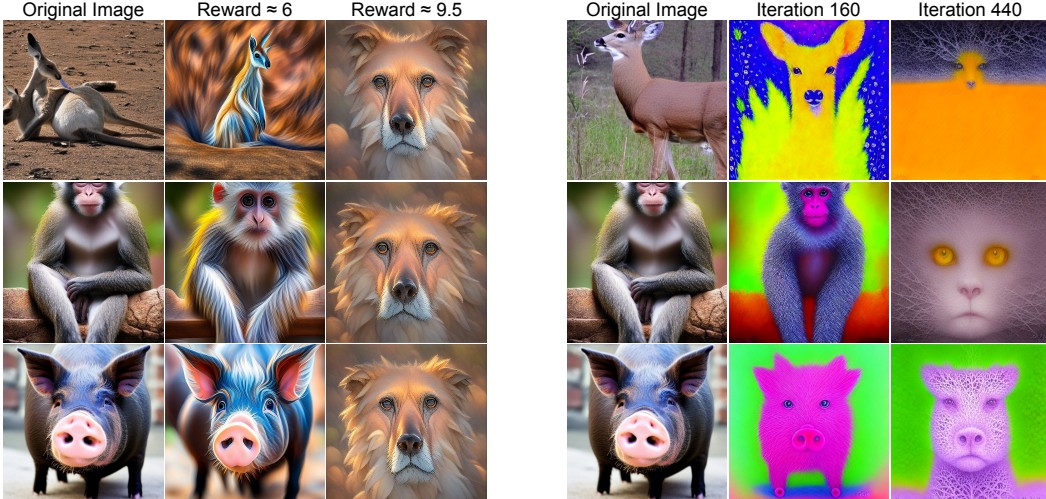

Figure 30: Reward hacking still occurs when we use large dropout rate in the Aesthetic classifier (Left). Incorporating a reward term to encourage diversity can lead to overly stylized images (Right).

### B.9 ROTATIONAL ANTI-CORRELATION REWARD

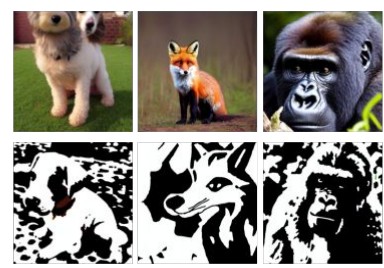

We also explored fine-tuning for symmetry or anti-symmetry properties. Here, we optimize for rotational anti-correlation, which encourages the diffusion model to generate images that are *dissimilar* to their rotations at $\{90, 180, 270\}$ degrees. Given an image $\mathbf{x}$ and rotation operator $\text{Rot}(\mathbf{x}, 90°)$, we formulate the objective as: $\mathbb{E}_{\mathbf{c} \sim p_{\mathbf{c}}, \mathbf{x} \sim \text{sample}(\boldsymbol{\theta}, \mathbf{c})} \left[ \frac{1}{3} \Big( \|\mathbf{x} - \text{Rot}(\mathbf{x}, 90°)\|^2 + \|\mathbf{x} - \text{Rot}(\mathbf{x}, 180°)\|^2 + \|\mathbf{x} - \text{Rot}(\mathbf{x}, 270°)\|^2 \Big) \right]$. The results are shown in Figure 31: this objective leads to interesting black-and-white, stylized images.

Figure 31: Rotational anti-correlation.

### B.10 CLIP REWARD

We briefly explored using $r(\mathbf{x}_0, \mathbf{c}) = \text{CLIP-Similarity}(\mathbf{x}_0, \mathbf{c})$ as a reward to improve image-text alignment. We found that optimization was succesful in that CLIP scores improved, but that image quality degraded after many steps of training (see Figure 32).

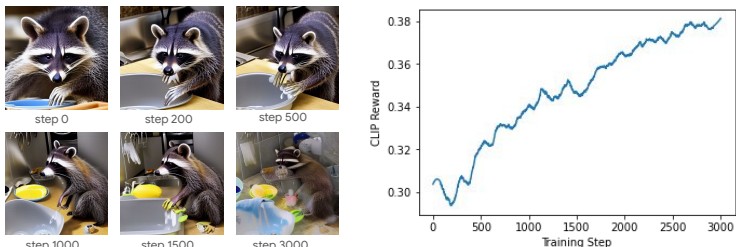

Figure 32: Qualitative examples (left) and training curve (right) for fine-tuning on CLIP reward using DRaFT for the prompt "a raccoon washing dishes".

Both HPSv2 and ImageReward are fine-tuned CLIP models, and for HPSv2 the annotators were specifically instructed to consider alignment to the prompt in their preference judgements. Therefore,

they already capture some aspects of image alignment, and qualitatively we did observe alignment to increase in some cases when using these rewards (e.g. in the "a raccoon washing dishes" example in Figure 1). However, we think improving text alignment using a powerful image captioning model such as PaLI (Chen et al., 2023) as the reward would be an interesting future direction.

### B.11 Fine-Tuning for Safety

The datasets on which large diffusion models are trained often contain NSFW content, which poses challenges for the deployment of safe models. Safety measures can be implemented at various stages of the text-to-image pipeline: 1) at the text prompt level (e.g., refusing NSFW words); 2) during the generative process (e.g., encouraging the model to output safe images); and 3) post-hoc elimination of NSFW images that have been generated. These approaches are complementary, and in practice we imagine that all three would be used in concert. Here, we focus on stage (2), by fine-tuning diffusion models using feedback from a pre-trained NSFW classifier. This approach is particularly useful for ambiguous prompts that may lead to either SFW or NSFW generations; for example, the prompt "swimsuit" can refer to images of people wearing swimsuits, or to just the article of clothing itself (the latter being more safe-for-work). We found that, before fine-tuning, the model mostly generated images of people wearing swimsuits, while after fine-tuning, the model instead generated images of lay-flat swimsuits.

### B.12 Learning Sampler Hyperparameters

As DRaFT (without truncated backprop through time) computes gradients through the denoising process end-to-end, we can use DRaFT to optimize sampler hyperparameters as well as model weights.

Learning sampler hyperparameters is similar in spirit to Watson et al. (2022), except we obtain gradients from a reward model rather than using kernel inception distance. We explored learning two inputs to the sampler—the guidance weights and negative prompt embeddings—although it would be possible to extend this method to other hyperparameters such as the time schedule. In particular, we investigated:

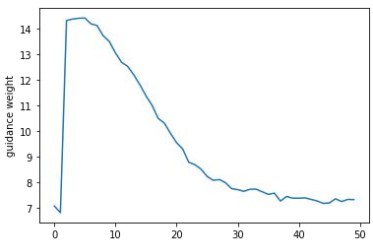

Figure 33: Guidance schedule learned by DRaFT

- **Guidance weights:** We learn a separate guidance weight $w(t)$ for each step of DDIM sampling. The learned schedule can be applied to samplers with different number of steps through linearly interpolating the schedule to stretch or compress it. We use a 50x larger learning rate for training guidance than the LoRA parameters because guidance weights have a large magnitude. An example learned guidance schedule is shown in Figure 33.

- **Negative prompt:** We learn prompt-dependent text embeddings passed into the unconditional UNet used for classifier-free guidance. In particular, we parameterize the embedding as MLP(Multi-Headed-Self-Attention($e$)) where $e$ is the text embeddings produced by the diffusion model's text encoder.

However, in practice, we found that this did not improve results much at large scale.

## C Method extensions

Here we briefly discuss two gradient-based reward fine-tuning methods we explored, but which we found did not achieve strong results compared to DRaFT-1 and DRaFT-LV.

### C.1 Single-Reward DRaFT-LV

A downside of DRaFT-LV is that it requires computing the reward function gradient an additional $n$ times. While this is relatively little overhead for the reward functions we used compared to the cost of image sampling, larger reward models could make this costly. We therefore experimented

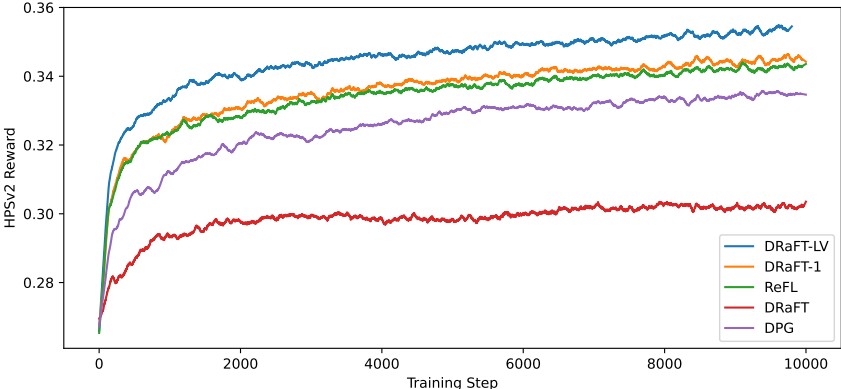

Figure 34: Training efficiency of reward fine-tuning methods on the HPSv2 reward.

with a version of DRaFT-LV that only computes the reward gradient with respect to the input pixels $\nabla_{\mathbf{x}} r(\mathbf{x}_0, \mathbf{c})$ once. Then it updates the gradient in the inner loop as $\boldsymbol{g} = \boldsymbol{g} + \nabla_{\mathbf{x}} r(\mathbf{x}_0, \mathbf{c}) \nabla_{\boldsymbol{\theta}} \hat{\mathbf{x}}_0$. Unfortunately, we found this method to be unstable and ultimately not learn effectively, possibly because it relies on the assumption that $\nabla_{\mathbf{x}} r(\mathbf{x}_0, \mathbf{c}) \approx \nabla_{\mathbf{x}} r(\hat{\mathbf{x}}_0, \mathbf{c})$, which may not be true in practice.

## C.2 DETERMINISTIC POLICY GRADIENT

Previous approaches applying RL to reward learning do not make use of sampling gradients. Here we discuss how Deterministic Policy Gradient (DPG; Silver et al. 2014; Lillicrap et al. 2016) offers a way of employing reward gradients within the RL framework.

Similarly to Black et al. (2023), we can view DDIM sampling as applying a deterministic policy to a deterministic Markov decision process:

- A state $\mathbf{s}_t$ consists of the current latent, timestep, and prompt $(\mathbf{x}_t, t, \mathbf{c})$.
- The initial state $\mathbf{s}_0$ draws $\mathbf{x}_T \sim \mathcal{N}(\mathbf{0}, \mathbf{I})$, $\mathbf{c} \sim p_{\mathbf{c}}$, and sets $t = T$.
- The policy is the learned denoiser: $\mathbf{a}_t = \boldsymbol{\mu}_{\boldsymbol{\theta}}(\mathbf{s}_t) = \boldsymbol{\epsilon}_{\boldsymbol{\theta}}(\mathbf{x}_t, \mathbf{c}, t)$
- The transition function performs a DDIM sampling step:
  $f(\mathbf{s}_t, \mathbf{a}_t) = (\frac{\alpha_{t-1}}{\alpha_t}(\mathbf{x}_t - \sigma_t \mathbf{a}_t) + \sigma_{t-1} \mathbf{a}_t, t-1, \mathbf{c})$
- The reward $R(\mathbf{s}_t, \mathbf{a}_t)$ is $r(\mathbf{x}_0, \mathbf{c})$ if $t = 0$ and 0 if otherwise.

DPG trains a critic $Q_{\boldsymbol{\phi}}(\mathbf{s}_t, \mathbf{a}_t)$ and learns $\boldsymbol{\theta}$ with the gradient update $\nabla_{\mathbf{a}} Q_{\boldsymbol{\phi}}(\mathbf{s}_t, \mathbf{a}_t) \nabla_{\boldsymbol{\theta}} \boldsymbol{\mu}_{\boldsymbol{\theta}}(\mathbf{s}_t)$ over sampled trajectories. The critic is trained to minimize the squared error between its prediction and the final return (in our case $r(\mathbf{x}_0, \mathbf{c})$).

Many of our rewards make use of neural networks with pre-trained parameters $\boldsymbol{\xi}$, which we denote by writing $r_{\boldsymbol{\xi}}$. We suggest applying LoRA to $\boldsymbol{\xi}$ parameterize the critic; we denote the LoRA-adapted parameters as $\text{adapt}(\boldsymbol{\xi}, \boldsymbol{\phi})$. We then apply the critic by (1) executing the action $\mathbf{a}_t$ on $\mathbf{s}_t$, (2) 1-step denoising the result to get a predicted clean image, and (3) applying the adapted reward model $r_{\text{adapt}(\boldsymbol{\xi}, \boldsymbol{\phi})}$ to the result. The reason for step (2) is to make the training of $\boldsymbol{\phi}$ more efficient: we believe it is easier to adapt the reward model parameters to produce good expected return estimates for one-step-denoised inputs than it is for noisy inputs. Formally, we use the critic:

$$Q_{\boldsymbol{\phi}}(\mathbf{s}_t, \mathbf{a}_t) = r_{\text{adapt}(\boldsymbol{\xi}, \boldsymbol{\phi})}((f(\mathbf{s}_t, \mathbf{a}_t) - \sigma_t \boldsymbol{\epsilon}_{\boldsymbol{\theta}}(f(\mathbf{s}_t, \mathbf{a}_t), \mathbf{c}, t-1))/\alpha_t, \mathbf{c}) \qquad (3)$$

When computing gradients through $Q$, we do not backpropagate through step (2) into $\boldsymbol{\theta}$, to prevent it from interfering with the policy training.

**Results.** The training efficiency of DPG compared with other methods is shown in Figure 34 (using the same experimental settings as in Figure 4). DPG outperforms vanilla DRaFT, but not our more-efficient variants. One challenge is that $Q$ produces poor return estimates for large timesteps, which

could perhaps be alleviated with a ReFL-style version that skips or rarely samples high timesteps for training the policy. There are many variants of DPG to explore, so we think that improving the performance of DPG for reward fine-tuning could be an interesting direction for future research; DPG is appealing because like vanilla DRaFT (but unlike ReFL and our DRaFT variants), it optimizes the full sampling process in an unbiased way.

## D  EXTENDED RELATED WORK

**Diffusion Models.**  Denoising diffusion probabilistic models (DDPMs; Sohl-Dickstein et al. 2015; Song & Ermon 2019; Song et al. 2021b; Ho et al. 2020) are a class of generative models that have become the de-facto standard for most continuous data modalities, including images (Ramesh et al., 2021), videos (Ho et al., 2022; Singer et al., 2023), audio (Liu et al., 2023), and 3D models (Zeng et al., 2022; Poole et al., 2023; Gu et al., 2023). Text-to-image diffusion models, which generate images conditioned on text prompts, have become prominent tools with the advent of models such as GLIDE (Nichol et al., 2021), DALLE-2 (Ramesh et al., 2022), Imagen (Saharia et al., 2022; Ho et al., 2022), and Latent Diffusion (Rombach et al., 2022). In this paper we focus on fine-tuning text-to-image models, but DRaFT is general and could be applied to other diffusion models or fine-tuning tasks, and to different domains such as audio or video.

**Learning human preferences.**  Human preference learning trains models on judgements of which behaviors people prefer, rather than on human demonstrations directly (Knox & Stone, 2009; Akrour et al., 2011). This training is commonly done by learning a reward model reflecting human preferences and then learning a policy that maximizes the reward (Christiano et al., 2017; Ibarz et al., 2018). Preference learning methods such as reinforcement learning from human feedback (RLHF) have become widely used for fine-tuning large language models so they improve at summarization (Stiennon et al., 2020; Wu et al., 2021), instruction following (Ouyang et al., 2022), or general dialogue (Askell et al., 2021; Bai et al., 2022a; Glaese et al., 2022; Bai et al., 2022b). We apply DRaFT to optimize scores from existing preference models, such as PickScore (Kirstain et al., 2023) and Human Preference Score v2 (Wu et al., 2023a), which are trained on human judgements between pairs of images generated by diffusion models for the same prompt.

**Guidance.**  Guidance steers the sampling process towards images that satisfy a desired objective by adding an auxiliary term to the score function. The canonical example is classifier guidance (Song et al., 2021b; Dhariwal & Nichol, 2021), which can turn an unconditional diffusion model into a class-conditional model by biasing the denoising process using gradients from a pre-trained classifier. Many other pre-trained models can also be used to provide guidance signals, including segmentation models, facial recognition models, and object detectors (Bansal et al., 2023). However, care must be taken when computing the guidance signal; naïvely, noisy examples obtained during the iterative refinement process may be out-of-distribution for the guidance model, leading to inaccurate gradients. Two approaches for mitigating this are (1) training a guidance model from scratch on noisy data (Dhariwal & Nichol, 2021) and (2) applying a pre-trained guidance model (which has seen only non-noisy data) to the one-step-denoised images obtained by removing the predicted noise at time $t$ (Li et al., 2022). However, both approaches have downsides: (1) precludes the use of off-the-shelf pre-trained guidance functions and (2) means the guidance is applied to out-of-distribution blurry images for high noise levels. DRaFT avoids these issues by backpropagating through sampling so the reward function is only applied to the fully denoised image.

**Backpropagation through diffusion sampling.**  Similarly to our method, Watson et al. (2022) backpropagate through sampling using the reparameterization trick and gradient checkpointing. They learn parametric few-step samplers that aim to reduce the inference cost while maintaining perceptual image quality, using a differentiable image quality score. Fan & Lee (2023) also propose an approach to speed up sampling: they consider doing this using gradients by backpropagating through diffusion sampling, but due to memory and exploding/vanishing gradient concerns, they focus on an RL-based approach. Both Fan & Lee (2023) and Watson et al. (2022) aim to improve sampling speed, rather than optimize arbitrary differentiable rewards, which is our focus. Like DRaFT, Direct Optimization of Diffusion Latents (DOODL; Wallace et al. 2023) uses backpropagation through sampling to improve image generation with respect to differentiable objectives. Rather than optimizing model parameters, DOODL optimizes the initial noise sample $\mathbf{x}_T$. Although DOODL does not require a training phase, it is much slower at inference time because the optimization must be redone for each prompt and metric. In contrast, after finetuning with DRaFT, sampling has the same cost as a standard

diffusion model. Furthermore, the latent optimization methods do not support the compositionality and interpolation properties that we can achieve with different sets of LoRA parameters tuned for different objectives.

Outside of image generation, Wang et al. (2023a) train diffusion model policies for reinforcement learning by backpropagating the Q-value function through sampling.

**Reward fine-tuning with supervised learning.**  Lee et al. (2023) and Wu et al. (2023b) use supervised approaches to fine-tune diffusion models on rewards. These methods generate images with the pre-trained model and then fine-tune on the images while weighting examples according to the reward function or discarding low-reward examples. Unlike DRaFT or RL methods, the model is not trained online on examples generated by the current policy. However, Dong et al. (2023) use an online version of this approach where examples are re-generated over multiple rounds of training, which can be viewed as a simple kind of reinforcement learning.

**Reward fine-tuning with reinforcement learning.**  Fan & Lee (2023) interpret the denoising process as a multi-step decision-making task and use policy gradient algorithms to fine-tune diffusion samplers. Building on it, Black et al. (2023) and Fan et al. (2023) use policy gradient algorithms to fine-tune diffusion models for arbitrary black-box objectives. Rather than optimizing model parameters, Hao et al. (2022) apply RL to improve the input prompts. RL approaches are flexible because they do not require differentiable rewards. However, in practice many reward functions are differentiable, or can be re-implemented in a differentiable way, and thus analytic gradients are often available. In such cases, using reinforcement learning discards useful information, leading to inefficient optimization.

**ReFL.**  Reward Feedback Learning (ReFL; Xu et al. 2023) evaluates the reward on the one-step predicted clean image, $r(\hat{\mathbf{x}}_0, \mathbf{c})$ from a randomly-chosen step $t$ along the denoising trajectory, and backpropagates through the one-step prediction w.r.t. the diffusion model parameters. DRaFT-$K$ is conceptually simpler than ReFL, as it only differentiates through the last few steps (e.g., $K = 1$) of sampling, where $K$ is deterministic; in (Xu et al., 2023), the authors randomly choose an iteration between a min and max step of the sampling chain (which incurs more hyperparameters) from which to predict the clean image. Also, because DRaFT runs the full sampling chain, our reward functions are always evaluated on final generations. In contrast, ReFL applies the rewards to one-step denoised latents in the sampling trajectory, similar to the use of one-step denoising in guidance. Furthermore, DRaFT-LV is substantially more efficient than ReFL, speeding up training by approximately $2\times$ by reducing the variance of the single-denoising-step gradient estimates.

# E    UNCURATED SAMPLES

Here, we show uncurated samples from models fine-tuned with DRaFT for different reward functions: Human Preference Score v2 (Figure 35), PickScore (Figure 36), and a combined reward (Figure 37). The first two are overfit to the reward functions, which decreases diversity and photorealism. In the third, we mitigate overfitting through LoRA scaling.

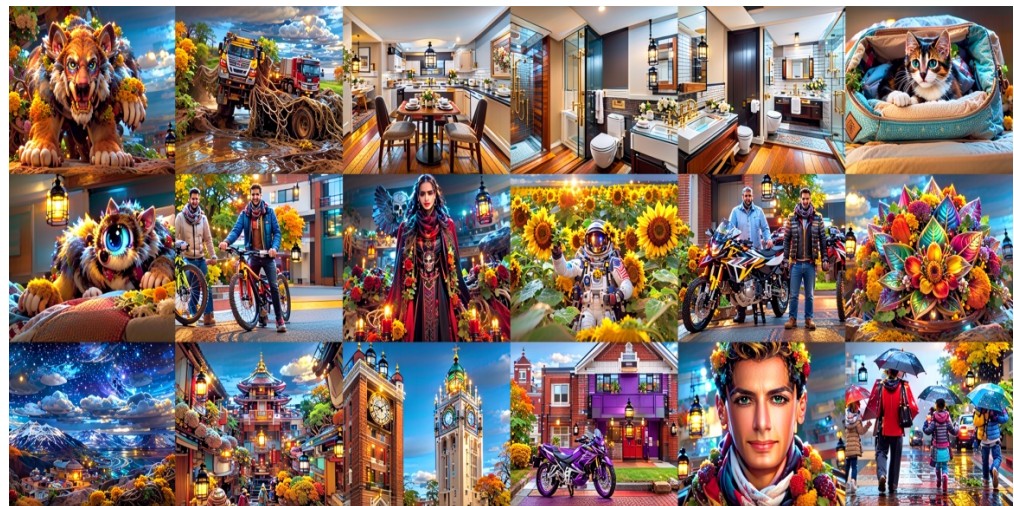

Figure 35: Uncurated samples from Stable Diffusion fine-tuned using DRaFT-1 for the HPSv2 reward (Wu et al., 2023a).

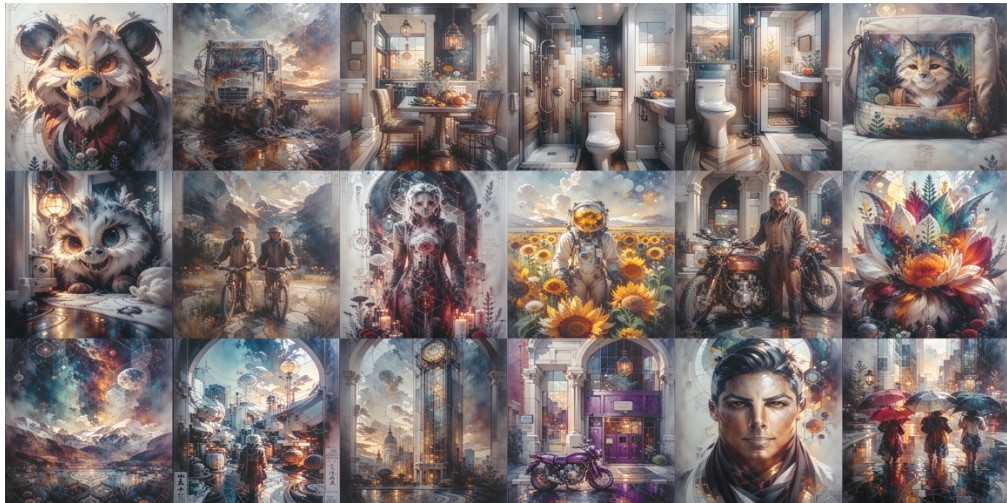

Figure 36: Uncurated samples from Stable Diffusion fine-tuned using DRaFT-1 for the PickScore reward (Kirstain et al., 2023).

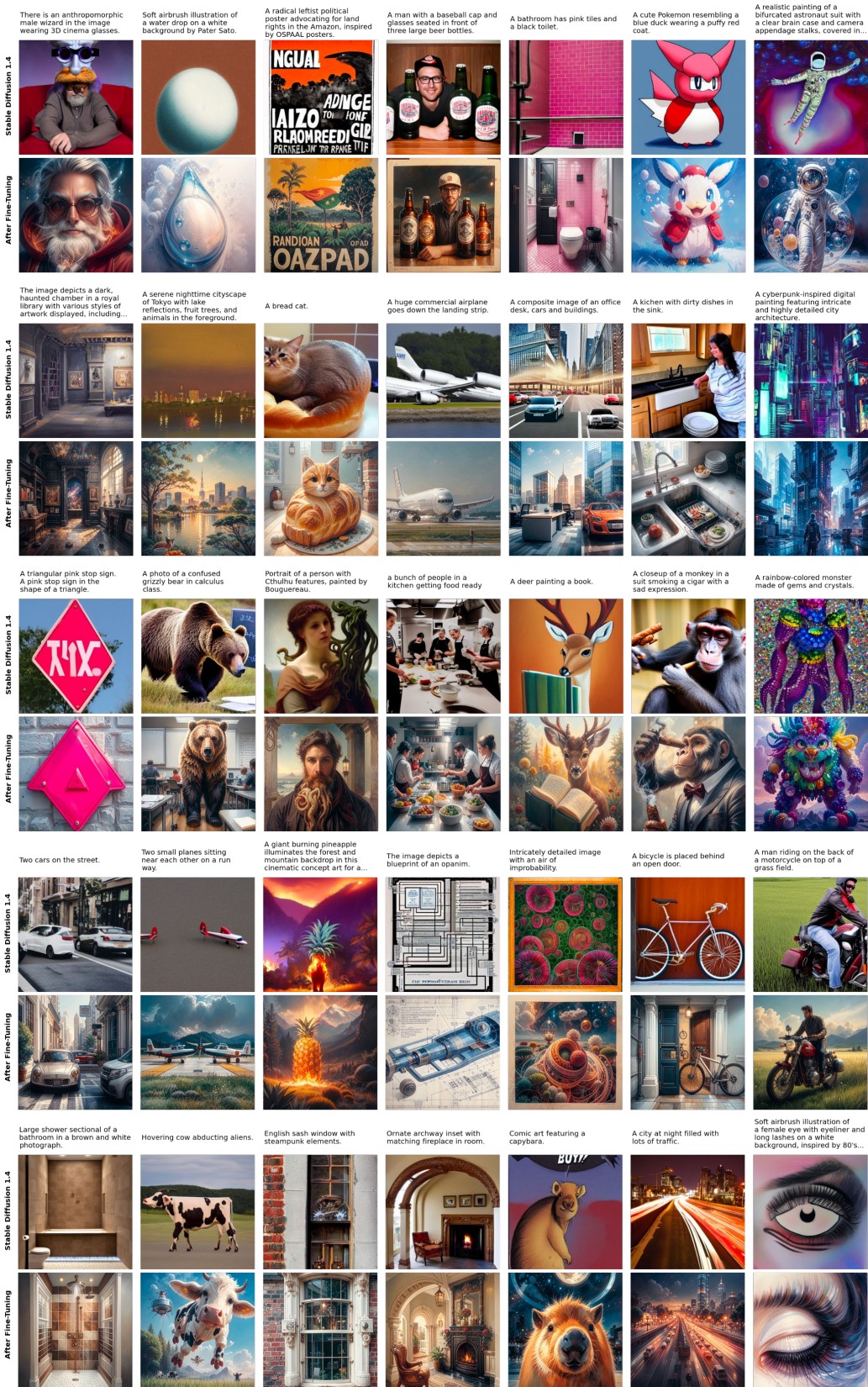

Figure 37: Uncurated samples from pre-trained Stable Diffusion 1.4 compared with DRaFT fine-tuned Stable Diffusion. We use DRaFT-LV on a combined reward of PickScore = 10, HPSv2 = 2, Aesthetic = 0.05. After training, LoRA weights are scaled down by a factor of 0.75 to reduce reward overfitting. Prompts are from the HPSv2 benchmark.

