# OpenReview forum: "Directly Fine-Tuning Diffusion Models on Differentiable Rewards"
_ICLR.cc/2024/Conference — ICLR 2024 poster_

### Official Review · Reviewer_tWnQ · 2023-11-01

**Soundness:** 4 excellent
**Presentation:** 4 excellent
**Contribution:** 2 fair
**Rating:** 5
**Confidence:** 4

**Summary:**

The paper propose Direct Reward Fine-Tuning (DRaFT) for finetuning diffusion models to maximize a differentiable reward functions. Several techniques are applied to circumvent large memory consumption and gradient exploding problem when they perform BPTT for diffusion models: gradient checkpointing, LoRA, ignoring early gradients. A variance-reduced alternative for DRaFT-1 is proposed. The authors provide solid evaluation on a vast set of differentiable rewards, with impressive qualitative results.

**Strengths:**

Optimizing a diffusion model with direct backprop through a differentiable reward has several technical difficulties and the authors managed to solve them.

(1) increased memory consumption. The authors solve the problem by combining two existing methods gradient checkpointing and LoRA.

(2) gradient exploding/vanishing problem for BPTT. The authors solve the problem by DRaFT-K, i.e., ignore gradients from earlier steps.

The base model is Stable-Diffusion v1.4. The authors include a vast range of differentiable rewards: (1) aesthetic classifier, (2) trained reward models from HPSv2, PickScore
(3) JPEG compressibility (4) Object Detector (5) ImageNet classifier. The experiments are solid and the qualitative results look impressive.

Several interesting observation:


1. compared to ReLF: it seems that DRaFT-1 (the same as ReFL with $m=1$) works better than ReLF with a random $m$.

2. Manipulating LoRA weights have predictable influence on the outputs.

**Weaknesses:**

The main concern is the significance of the contribution:

1. DRaFT-1 works the best but the improvement over ReFL is a bit marginal (Figs 3 and 4).  Furthermore, DRaFT-1 is the same as ReFL with $m$ fixed to 1. Increasing $K$ does not improve the performance.

2. As DRaFT-1 (and LV version) already works well, in this case, we might not need gradient checkpointing and LoRA, right? How does finetuning the full parameter influence the result?

**Questions:**

This is a bit minor, but why do you choose Stable-Diffusion v1.4 instead of SD v1.5 or a higher version?

---

> ### Author Response · Authors · 2023-11-21
> **Author response**
>
> Thank you for the helpful comments and suggestions. We address your questions and concerns below:
>
> **Q: DRaFT-1 works the best but the improvement over ReFL is a bit marginal (Figs 3 and 4).**
>
> **A:** Please see the general response: DRaFT-LV is 2x more efficient than ReFL, leading to higher rewards. Fast learning may be particularly important for future reward fine-tuning using large reward models or on-the-fly learning to a specific user’s preferences.
>
>
> **Q: Furthermore, DRaFT-1 is the same as ReFL with K fixed to 1. Increasing K does not improve the performance.**
>
> **A:** One of our contributions is unifying gradient-based approaches for diffusion fine-tuning in a common framework, as shown in Algorithm 1. The fact that DRaFT is equivalent to ReFL when $K=1$ provides a useful connection between the approaches, which we hope helps to understand the space of possible approaches. In addition, we show that using $K=1$ works surprisingly well and yields a conceptually simpler and easier-to-implement approach than ReFL, which randomly samples $K$ in a range specified by two hyperparameters (the min and max step number from which to predict a one-step denoised image). Please see the general response for further discussion about ReFL.
>
> **Q: As DRaFT-1 (and LV version) already works well, in this case, we might not need gradient checkpointing and LoRA, right? How does finetuning the full parameter influence the result?**
>
> **A:** Yes, you are correct that for small $K$, we do not need gradient checkpointing; we have clarified this in the paper (on page 4). Fine-tuning the full model parameters performs comparably to using LoRA, but LoRA substantially reduces memory cost. Additionally, because all of the information incorporated during fine-tuning is contained within the LoRA parameters, we can more easily store and combine a set of LoRA weights adapted for different rewards and use them to obtain semantic interpolations as shown in Figure 7.
>
>
> **Q: Why do you choose Stable-Diffusion v1.4 instead of SD v1.5 or a higher version?**
>
> **A:** We used Stable Diffusion 1.4 to have direct comparisons with DDPO.
>
>
> Thank you again for taking the time to read our paper and for providing thoughtful feedback. Please let us know if you have any other questions, and we would be happy to answer them.

---

### Official Review · Reviewer_m5FM · 2023-11-01

**Soundness:** 4 excellent
**Presentation:** 4 excellent
**Contribution:** 3 good
**Rating:** 8
**Confidence:** 4

**Summary:**

The paper introduces "Direct Reward Fine-Tuning (DRaFT)", as an efficient framework for fine-tuning diffusion models on differentiable rewards. As opposed to previous work, they develop tools to differentiate through the entire diffusion process and use the resulting gradients for fine-tuning. However, they find that the truncated versions of their method which only differentiate through the last time step (similar to previous work such as ReFL(Xu. et al 2023)) works the best. They introduce a variant of this method which computes a lower variance last step gradient by averaging across multiple noise samples. They show that this variant outperforms all baselines and show various interesting use cases of their method.

**Strengths:**

1. I think the biggest strength of this paper lies in the exhaustive ablations and use cases the authors show.
   - I particularly liked the experimentation with interpolating and scaling different LORA parameters and the corresponding effects.
   - I also appreciated the comparisons of the different reward functions and the qualitative differences between the model fine-tuned with them.
   - I also found some of the applications of this method very interesting, such as the object removal/inclusion example in section 5.3
   - Lastly, I also liked the reward hacking examples presented in the paper.
2) The paper is also very well written and was a pleasure to read.
3) The related work description was thorough and I enjoyed (and learnt from) reading the descriptions and comparisons.

**Weaknesses:**

The main methodological contributions of the paper seem to be the usage of LoRA and gradient checkpointing techniques to allow differentiating through the full diffusion chain. However, the paper shows that differentiating through the diffusion chain leads to exploding gradients without providing sufficient analysis of it beyond Fig 8 or any attempts to mitigate the issues. Instead most of the benefits in the results are attributed to DRaFT-LV which is (methodologically) a very minor improvement over ReFL. Although the usage of LoRA for fine-tuning even in this single step setting provides interesting possibilities (which the authors exploit rather well in the experiments), I feel the paper would have benefited from additional analysis and fixes for the full DRaFT setting as well.

**Questions:**

I have a broader philosophical question about the approach. I find the broader approach taken by the paper (and previous work) a little perplexing. For example, the paper at various points shows the perils of over-optimizing a specific objective due to effects like reward hacking. An obvious solution I would assume to a problem like this would be, using a dataset of (prompt, image) pairs with the image as x_0 in an algorithm like DRaFT-LV instead of sampling it using the diffusion process? or perhaps augmenting this as an auxiliary objective along with the regular DRaFT-LV objective? This seems 1) faster and 2) less susceptible to issues like reward hacking etc, as the model stays grounded to remain close to the real image distribution. I would be curious if the authors have experimented with similar methods?
- Other Minor comments:
    1) As a general comment, it would be helpful if all the generated images shown in the paper also mention the prompt used. There are
certain figures where I couldn’t spot the prompt used to generate the image. Eg. Fig 7.
    2) Further, in Fig 7, left, it would help if the paper used the same prompt for all 3 examples to contrast how the interpolations vary across the models.

---

> ### Author Response · Authors · 2023-11-21
> **Author response**
>
> Thans for the helpful comments and suggestions. We address your questions and concerns below:
>
> **Q: Investigating backpropagation through sampling**
>
> **A:** We provide an analysis of gradient norms, the impact of $K$, and the impact of deactivating LoRA parameters for different ranges of steps in Figures 8-10. We also note that DRaFT-$K$ is an approach to mitigate the issues of full backpropagation.
>
> Additionally, we used gradient clipping throughout our experiments. We have added an ablation over the gradient clipping norm hyperparameter in Appendix B.7 (see Figure 28). We compare the reward values achieved over the course of fine-tuning with gradient clipping norms $c \in \{ 0.001, 0.01, 0.1, 1.0, 10.0, 100.0, 1000.0 \}$. We found that, when using DRaFT-50, smaller gradient clipping norms ($c = 0.001$) improved optimization substantially, while large clipping norms ($c = 1000$) impeded training. This further supports the observation that backpropagating through long diffusion sampling chains can lead to exploding gradients. We performed a similar ablation for DRaFT-1, and found that the fine-tuning performance was nearly identical for all gradient clipping coefficients.
>
> Empirically, while vanilla DRaFT works better than RL, it is much less efficient than DRaFT-$K$ with small $K$, and thus we focused on DRaFT-$K$. However, we agree that further understanding the gradient flow through long sampling chains is an interesting direction for future work.
>
> **Q: Dealing with over-optimization and reward hacking**
>
> **A:** We agree that mitigating reward overfitting is an important issue, so we have added new experiments on this topic to Appendix B.8, Figure 29. We compared three strategies for mitigating reward hacking: early stopping, KL regularization towards the pre-trained model, and our LoRA scaling approach (discussed in Section 5.2 of the main text). We found LoRA scaling to be the most effective.
>
> We view optimizing reward functions and mitigating overfitting as separate goals that can be studied independently, and primarily focus on the former in this paper. As we mention in Section 5.1, we believe that efficient methods like DRaFT could be a key stepping stone towards designing improved rewards (or regularizers), as DRaFT allows one to quickly evaluate the effect of “over-optimizing” a reward.
>
> **Q: Suggested methods for mitigating reward overfitting.**
>
> **A:** Interesting suggestions! We run have run several experiments on this:
>
> **KL Regularization.** We agree that regularizing the model to stay close to the real image distribution is a good idea to reduce reward hacking. We experimented with one approach in this direction: adding KL regularization towards the pre-trained model. The pre-trained model implicitly captures the training image distribution, and this kind of regularization is standard practice for avoiding reward overfitting when performing RLHF with LLMs. We found that this approach helps mitigate reward overfitting (see Figure 29 in Appendix B.8), although it did not appear to work as well as the LoRA scaling approach we discuss in Section 5.2.
>
> **$\mathbf{x}_0$ Idea.** We also experimented with using fixed $\mathbf{x}_0$s with DRaFT-LV rather than ones generated online by the current policy. Counterintuitively, this approach actually makes reward overfitting worse and leads to poor results (0.333 HPSv2, 0.216 PickScore in the same setting as Figure 29). The main issue is with distribution shift: the latents produced during sampling at test time will not match the ones seen in training. In particular, the model has no way to correct itself if it starts following the reward model’s score function too strongly, which results in artifacts in the samples. Previous work has also found offline policy learning to not be as effective as using policy-generated images; for example, see the comparison between DDPO and RWR in arxiv.org/abs/2305.13301.
>
> **Q: Prompts for generated images**
>
> **A:** Thank you for the suggestion! For Figure 7, we used prompts from the HPDv2 dataset, which are quite long and hard to fit due to space constraints; Figures 19 and 20 in Appendix B.5 provide expanded visualizations that include the prompt on the left-hand side. We will also add more prompt text to the main paper, where space allows.
>
>
> **Q: Prompts for Figure 7.**
>
> **A:** Figures 19 and 20 in Appendix B.5 show the samples generated for the same set of prompts when fine-tuning for HPSv2 and PickScore, respectively. We felt it was useful to show a diversity of samples in the main paper (Figure 7). Also, Figure 5 in the main paper compares the effects of different reward functions on the same prompts (“a photo of a deer” and “a painting of a deer”).
>
> Thank you again for taking the time to read our paper and for providing thoughtful feedback. Please let us know if you have any other questions, and we would be happy to answer them.

---

> ### Comment · Reviewer_m5FM · 2023-11-22
> **Official comment by Reviewer m5FM**
>
> I thank the authors for the detailed reponse and for adding experiments related to gradient clipping showing the promise of gradient surgery to improve the optimizer performance. I also thank the authors for adding results related to mitigating reward hacking. Given the promise shown by a relatively simple fix on mitigating exploding gradients, I would have appreciated a more thorough treatment of the mechanisms involved and possible solutions to fix the exploding gradients issue. However, I understand the paper offers computational tools for future work to explore and understand those issues better. Thus, I'm bumping up my score to 7.
>
> Edit : Oops, just realized there is no 7. So making it 8 instead with some hesitation.

---

> > ### Author Response · Authors · 2023-11-23
> >
> > Thank you again for your thoughtful review. We believe our paper has been improved by incorporating your feedback, and we appreciate that you raised your score.

---

### Official Review · Reviewer_pZzS · 2023-11-02

**Soundness:** 3 good
**Presentation:** 3 good
**Contribution:** 3 good
**Rating:** 8
**Confidence:** 4

**Summary:**

This paper is another addition to a line of work which fine-tunes diffusion models on reward functions to generate samples with desirable properties. The closest work is ReFL, which backpropagates gradients through a differentiable reward function for a single-step denoising operation. The proposed method, DRaFT, calculates gradients for the entire diffusion sampling chain, which is facilitated by using two existing techniques: low-rank adaption (LoRA) and gradient checkpointing. Different variations of DRaFT are proposed to improve efficiency, reduce memory footprint and reduce variance of the gradient estimates. Experiments use Stable Diffusion 1.4 as the backbone and consider several reward functions such as the LAION aesthetic score, Human Preference Score v2 and PickScore.

**Strengths:**

This is primarily an empirical contribution. While the novelty on the algorithmic side is somewhat limited, this paper applies the idea to large pre-trained models and demonstrates effective performance on generation of realistic images. In terms of significance, this work is an improvement over existing stable diffusion models, with the benefit that using different reward functions enables fine-tuning for different types of characteristics in the generated samples.

The experiments are comprehensive and compare DRaFT against relevant baselines. The results showcasing interpolation between pre-trained and fine-tuned weights, as well as between LoRA weights for different reward functions are interesting and insightful. The various ablation studies and considering specific problems such as compression, object detection and adversarial examples demonstrate the wide applicability of this method.

**Weaknesses:**

The relatively minor weakness is related to the organization of the paper to improve clarity. Figure 2 can be moved in Section 4 to match the text with the corresponding model diagram. Section 5.2 discusses the results of many different experiments and while they are organized by paragraphs, splitting them into sub-sections with informative titles, or perhaps a description of the contents of Section 5 (similar to the one for the Appendix) can help navigate this section more easily.

**Questions:**

None

**Details Of Ethics Concerns:**

Since the backbone of this work is Stable Diffusion, the ethical concerns associated with such models also apply to this work.
Particularly, the training data for such models can use copyrighted images, and may violate the privacy of users in case the images contain humans. Generative models may also suffer from bias in their sampling procedures (usually a result of bias in the training data) and adequate measures must be taken to ensure fairness and safety when using such models.

---

> ### Author Response · Authors · 2023-11-21
> **Author response**
>
> Thank you for the helpful comments and suggestions. We address your questions and concerns below:
>
> **Q: Paper organization**
>
> **A:** Thank you for the suggestions for improving the organization of the paper! We used paragraphs rather than subsections due to space constraints. We have moved Figure 2 to Section 4 in the updated paper.
>
> Thank you again for taking the time to read our paper and for providing thoughtful feedback. Please let us know if you have any other questions, and we would be happy to answer them.

---

> > ### Comment · Reviewer_pZzS · 2023-11-22
> >
> > Thank you for the response!

---

> > > ### Author Response · Authors · 2023-11-23
> > >
> > > Thank you again for your thoughtful feedback!

---

### Official Review · Reviewer_LgKe · 2023-11-03

**Soundness:** 3 good
**Presentation:** 3 good
**Contribution:** 2 fair
**Rating:** 3
**Confidence:** 5

**Summary:**

The paper presents DRaFT for optimizing diffusion models using differentiable reward functions. While the direct fine-tuning idea has been around, the memory issue has been a bottleneck to scale this idea to a large-scale diffusion model. DRaFT uses LoRA/grad checkpointing to solve this problem. The authors propose show that DRaFT can successfully finetune image generative models, better than a reinforcement learning-based approach.

**Strengths:**

The paper presents an original approach to fine-tuning diffusion models using differentiable reward functions. The proposed method, DRaFT, and its variants, DRaFT-K and DRaFT-LV, demonstrate strong performance across a variety of reward functions, outperforming one reinforcement learning-based method. The paper is generally well-written and clear, and the experimental results are promising. The significance of the work lies in its potential to enhance the aesthetic quality of images generated by diffusion models, a valuable contribution to the field of generative models.

**Weaknesses:**

The paper lacks a thorough discussion of related work.
1) “Black et al. (2023) and Fan et al. (2023) interpret the denoising process as a multi-step decision-making task, and use policy gradient algorithms to fine-tune diffusion models for arbitrary black-box objectives.” => This interpretation is originally from [1].
2) the idea of directly fine-tuning diffusion models on differentiable rewards is also studied in [1], but the authors do not seem to be aware of it. (See Sec 3.3)

Also, I wonder if other PEFT methods such as Adapters, Normalization training, and IA3 could be used instead of LoRA.

(very minor) The use of the "stop_grad" function in the pseudocode is not properly explained, which could lead to confusion for readers unfamiliar with this function.

[1] Reinforcement learning for faster DDPM sampling

**Questions:**

- Could you elaborate on the relationship between your work and that of [1], especially regarding the idea of directly fine-tuning diffusion models on differentiable rewards?
- Can DRaFT work with other PEFT methods such as Adapters, Normalization training, and IA3?
- The paper does not report the performance of DPOK. Could you clarify how your approach compares to this method?

---

> ### Author Response · Authors · 2023-11-21
> **Author response**
>
> Thank you for your helpful comments and suggestions. We address your questions and concerns below:
>
> **Q: The paper lacks a thorough discussion of related work.**
>
> **A:** Our related work section (Section 2) covers: 1) learning human preferences, 2) guidance, 3) backpropagation through diffusion sampling, 4) reward fine-tuning with supervised learning, 5) fine-tuning with RL, and 6) reward feedback learning. In our original submission, we also included an extended discussion of related work in Appendix D. We have updated the main text to cite and discuss the paper you pointed out.
>
> **Q: Could you elaborate on the relationship between your work and that of [1], especially regarding the idea of directly fine-tuning diffusion models on differentiable rewards?**
>
> **A:** We could not find a paper with the title you referenced, “Reinforcement learning for faster DDPM sampling.” The closest paper we found was Fan & Lee, “Optimizing DDPM Sampling with Shortcut Fine-Tuning,” ICML 2023. Is this the paper you are referring to? If so, then we note that its focus is different from our work. Fan & Lee (2023) propose an approach to speed up sampling: they consider the possibility of doing this using gradients by backpropagating through diffusion sampling, but due to memory and exploding/vanishing gradient concerns, they focus on an RL-based approach.
>
> We propose a thorough framework for gradient-based diffusion fine-tuning that addresses the memory concern by leveraging LoRA parameters and gradient checkpointing, and addresses exploding gradients by using truncated backpropagation through sampling. Then, we show empirically that the method works well over a diverse set of challenging tasks that could not previously be optimized efficiently with RL-based approaches. Additionally, we propose several modifications (DRaFT-LV, LoRA mixing) that further improve performance over the base approach.
>
> The idea of backpropagating through sample quality was studied empirically by Watson et al. (2022) (https://arxiv.org/abs/2202.05830), which we discuss in the paper. Importantly, the goals of Fan & Lee (2023) and Watson et al. (2022) are very different from ours: they focus on improving sampling speed, whereas we focus on optimizing arbitrary differentiable rewards. That being said, we agree that Fan & Lee (2023) is a very relevant prior work and we have added a citation and discussion of it in our related work (Section 2, in both the "backpropagation through diffusion sampling" and "reward fine-tuning with reinforcement learning" paragraphs). Thank you for pointing it out.
>
>
> **Q: Diffusion as a multi-step decision-making task.**
>
> **A:** Thank you for pointing out the Fan & Lee (2023) paper that was the first to interpret denoising as a multi-step decision-making task. We have updated the related work section to cite this paper (Section 2, in the paragraph on reinforcement learning).
>
> **Q: Can DRaFT work with other PEFT methods such as Adapters, Normalization training, and IA3?**
>
> **A:** It certainly could and this is a great direction to pursue, although we leave exploring that direction for future work.
>
>
> **Q: The paper does not report the performance of DPOK. Could you clarify how your approach compares to this method?**
>
> **A:** DPOK is an RL approach similar to DDPO (and done concurrently). While there are some differences (e.g. adding KL regularization but not using an importance sampling estimator to perform multiple updates), we expect that it will perform similarly to DDPO. In contrast, DRaFT learns 200x faster than DDPO because it leverages reward gradient information (see Figure 3).
>
>
> **Q: The use of the "stop_grad" function in the pseudocode is not properly explained**
>
> **A:** We explain the use of `stop_grad` in the “General Reward Fine-Tuning Algorithm” paragraph in Section 4. We have further clarified the role of `stop_grad` in the updated text. Thank you for the suggestion.
>
> Thank you again for taking the time to read our paper and for providing thoughtful feedback. Please let us know if you have any other questions, and we would be happy to answer them.

---

### Author Response · Authors · 2023-11-21
**General response to reviewers**

We thank all the reviewers for their thoughtful comments! We respond to each reviewer individually, and have updated the paper to take into account the reviewers’ suggestions (changes are highlighted in green). We would also like to address the comparison with ReFL here, which was brought up by reviewers tWnQ and m5FM. We believe that our paper has important contributions compared to ReFL, with regard to both novelty and results.
* Empirically, we show that DRaFT-LV learns 2x faster than ReFL and achieves higher rewards (see Figures 3 and 34). Additionally, we show that scaling DRaFT-LV to using multiple rewards and then regularizing it with LoRA scaling is an effective recipe for preference fine-tuning that can substantially improve sample quality.
* Methodologically, we present a general reward fine-tuning framework that includes ReFL as a special case, but also covers multiple alternatives and improvements.
* Beyond the method, we believe the paper contains substantial contributions that will be valuable to the community, including studying numerous reward functions, evaluating generalization, and mixing LoRA parameters adapted for different rewards.

---

### Meta-Review · Area_Chair_L54Z · 2023-12-11

**Metareview:**

**Paper Summary**: The paper introduces Direct Reward Fine-Tuning (DRaFT), a method for fine-tuning diffusion models to maximize differentiable reward functions, like human preference scores. DRaFT addresses the challenge of backpropagating through the entire diffusion sampling process. Variants like DRaFT-K and DRaFT-LV are proposed to improve efficiency and reduce variance. DRaFT demonstrates superior performance over reinforcement learning-based methods and can significantly improve the aesthetic quality of images generated by Stable Diffusion 1.4. The paper also explores the use of DRaFT with various reward functions and presents a unifying perspective on gradient-based fine-tuning algorithms.

**Strengths**: The primary strength of the paper is its novel approach to fine-tuning diffusion models using differentiable reward functions. DRaFT and its variants overcome technical challenges, such as increased memory consumption and gradient exploding/vanishing issues, by using gradient checkpointing and LoRA. The method's effectiveness is demonstrated through comprehensive experiments, showcasing improvements in image generation quality. The approach is versatile, as it adapts to various reward functions, and offers potential improvements over existing stable diffusion models.

**Weaknesses**: The main weakness of the paper is its focus on DRaFT-LV, which is a marginal improvement over existing methods like ReFL. The methodological contributions are somewhat limited, and there's insufficient analysis of exploding gradients in the full DRaFT setting. While the paper showcases various applications and ablations, it lacks deeper exploration and potential solutions for the challenges of backpropagating through the entire diffusion chain. Additionally, the organization of the paper could be improved for clarity, and there's a need for more thorough comparisons with similar methods.

**Justification For Why Not Higher Score:**

After all discussion, the reviewers still expressed a desire for a more thorough treatment of exploding gradients and other mechanisms.

**Justification For Why Not Lower Score:**

Reviewers appreciated the authors' detailed response and the added experiments addressing gradient clipping, which showed promise in improving optimizer performance. The authors' additional results on mitigating reward hacking were also acknowledged. Despite these points, reviewers were persuaded by the authors' arguments and implementations, leading to an increase in the overall evaluation score of the paper.

---

### Decision · Program_Chairs · 2024-01-16

Accept (poster)